# RNNs with gracefully degrading continuous attractors

## Abstract

Attractor networks are essential theoretical components in recurrent networks for memory, learning, and computation. However, the continuous attractors that are essential for continuous-valued memory suffer from structural instability—infinitesimal changes in the parameters can destroy the continuous attractor. Moreover, the perturbed system's dynamics can exhibit divergent behavior with associated exploding gradients. This poses a question about the utility of continuous attractors for systems that learn using gradient signals. To address this issue, we use Fenichel's persistence theorem from dynamical systems theory to show that bounded attractors are stable in the sense that all perturbations maintain the stability. This ensures that if there is a restorative learning signal, there will be no exploding gradients for any length of time for backpropagation. In contrast, unbounded attractors may devolve into divergent systems under certain perturbations, leading to exploding gradients. This insight also suggests that there can exist homeostatic mechanisms for certain implementations of continuous attractors that maintain the structure of the attractor sufficiently for the neural computation it is used in. Finally, we verify in a simple continuous attractor that all perturbations preserve the invariant manifold and demonstrate the principle numerically in ring attractor systems.

## 1 Introduction

Recurrent neural/neuronal networks (RNNs) can process sequential observations and model temporal dependencies of arbitrary length. At the same time, they are fundamentally limited by their finite-sized hidden states which form the only channel between the past and the future. To store information over a long period of time, as many difficult tasks demand, RNNs can learn or be designed to have "persistent memory". When the information of interest is continuous-valued, a natural solution is to use continuous attractors. Continuous attractors are prevalent in theoretical neuroscience as tools to model neural representation and computation ranging from internal representations of head directions and eye positions to perceptual decision-making and working memory (Khona and Fiete, 2022). Continuous attractors are also at the core of long short-term memory (LSTM) (Greff et al., 2017) units and the neural Turning machine (NTM) (Graves et al., 2014) to provide digital computer memory like properties not natural to recurrent networks. In fact, the critical weakness of continuous attractors is their inherent brittleness as they are rare in the parameter space, i.e., infinitesimal changes in parameters destroys the continuous attractor structures implemented in RNNs (Seung, 1996; Renart et al., 2003), even if biologically plausible asymmetric connections are used to construct them (Darshan and Rivkind, 2022). However, as we will show, not all RNN implementations of continuous attractors behave similarly in their brittleness.

We found that in the space of RNNs, some have neighbourhoods with highly undesirable exploding gradients. We will describe some continuous attractors which have such neighbourhoods. Consider an RNN (without input or output for now) expressed in continuous time as an ordinary differential equation:

$$\dot{\mathbf{x}} = -\mathbf{x} + [\mathbf{W}\mathbf{x} + \mathbf{b}]_+ \tag{1}$$

where $\mathbf{x} \in \mathbb{R}^d$ is the hidden state of the network, $\mathbf{b} > 0$ is the bias, and $[\cdot]_+ = \max(0, \cdot)$ is the threshold nonlinearity per unit. In discrete time, this corresponds to a ReLU RNN (see Sec. 3.1.1). The non-trivial activity of this network is limited to the (non-negative) first quadrant.

When $d = 2$, we can build two kinds of continuous attractors. First, through positive feedback, $\mathbf{W} = [0, 1; 1, 0]$ and no bias $\mathbf{b} = \mathbf{0}$, we can create a continuous attractor, i.e., $\dot{\mathbf{x}} = 0$ on the $x_1 = x_2 \geq 0$ half-line, and surrounding attractive flow (Fig. 1A left). We refer to it as an **unbounded line attractor (UBLA)**. For any point on the line attractor, linearization results in eigenvalues $0$ and $-2$, corresponding to the zero flow and attractive flow respectively. When $\mathbf{W}$ is perturbed, the null eigenvalue can easily become non-zero and the continuous line attractor disappears. If it becomes negative, the system bifurcates to a stable fixed point at the origin (Fig. 1A bottom). However, if it becomes positive (Fig. 1A top), *the resulting flow diverges to infinity along the diagonal*. Corresponding to the divergent flow, the backpropagating gradient over time exponentially grows in magnitude, thus rendering gradient descent impractical without truncation in time.

The second kind of continuous attractor is created through negative feedback. By choosing $\mathbf{W} = [0, -1; -1, 0]$ and $\mathbf{b} = [1; 1]$, we get $\dot{\mathbf{x}} = 0$ on the $x_1 = -x_2 + 1$ line segment in the first quadrant as the continuous attractor. We refer to it as the **bounded line attractor (BLA)**. Again, linearization on the attractor shows two eigenvalues, $0$ and $-2$, and perturbations again cause the null eigenvalue to be non-zero and the line attractor disappears. However, surprisingly, the bifurcations are qualitatively different. It either bifurcates into a single stable fixed point (Fig. 1B top) or two stable fixed points separated with a saddle node in between (Fig. 1B bottom). Neither of these two cases show a divergent flow, but rather consists of one or two basins of attraction. It implies only vanishing gradients for this system and **exploding gradients will not be present for an arbitrarily long time**.

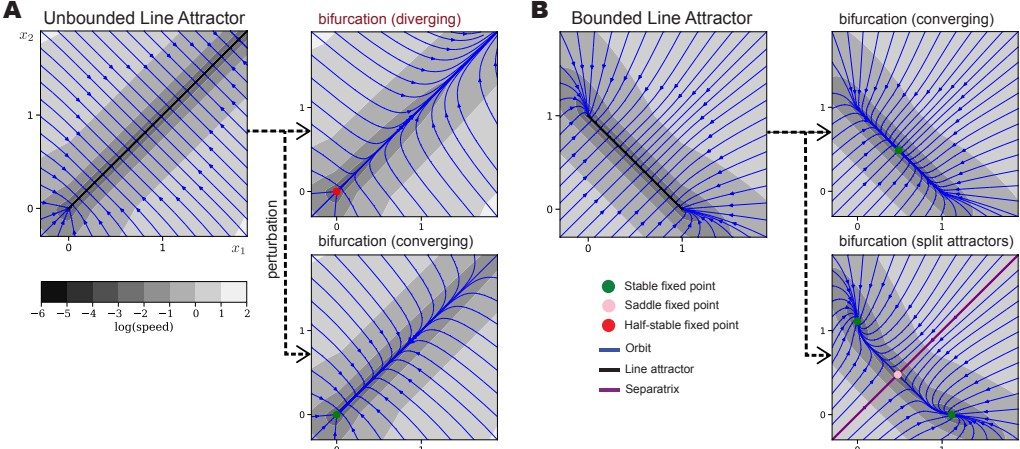

Figure 1: Motivating case study of the two systems implementing the same computation but one near exploding gradients. Phase portraits for unbounded and bounded linear attractors (1). Under perturbation of parameters, each of them can bifurcate to one of the two potential systems without the continuous line attractor. Note that the parameters for the UBLA are near a diverging system associated with exploding gradient behavior.

Avoiding exploding gradients is of importance in the context of long-range temporal learning (Sec. 2.2). Learning generally induces stochasticity in parameters, while spontaneous synaptic fluctuations are present in biological neuronal networks (Sec. 2.3). Given these observations, we predict that BLA would be a more stable motif for computation than UBLA in the presence of noise and continuous learning. Since BLAs, but not UBLAs, avoid exploding gradients, if the desired computation requires a line attractor of finite range, BLA would be both easier to maintain and learn. Is this only true for ReLU parameterized RNNs, or does it generalize?

In this paper, we lay out a new theory of general continuous-valued memory in the context of learning to answer the following questions:

1. Can we avoid exploding gradients under parameter perturbation?
2. Do we need to worry about the brittleness of the continuous attractor solutions in practice?

Our theory provides answers to both questions under mild assumptions in an architecture agnostic manner. Using Fenichel's invariant manifold theorem, we derive a sufficient condition for RNNs

implementing continuous attractors to remain free of exploding gradients. Moreover, even after a bifurcation, these RNNs still approximately behave like the original continuous attractor for a while. Together these theoretical results significantly mitigate the concern of the fine tuning problem in theoretical neuroscience and suggest general principles for evaluating and designing new architectures and initialization strategies for RNNs in machine learning.

## 2 THEORY OF GRACEFULLY DEGRADING CONTINUOUS ATTRACTORS

In this section, we apply invariant manifold theory to RNNs and translate the results for the machine learning and theoretical neuroscience audience. Our emphasis in this paper centers on investigating the distinctive properties of continuous attractors that prove essential for specific tasks, with a deliberate exclusion of considerations related to learning chaotic dynamics.

### 2.1 INVARIANT CONTINUOUS ATTRACTOR MANIFOLD THEORY

We start by formulating RNNs implementing a continuous attractor in continuous time: $\dot{\mathbf{x}} = \mathbf{f}(\mathbf{x})$. Let $l$ be the intrinsic dimension of the manifold of equilibria that defines the continuous attractor. We will reparameterize the dynamics around the manifold with coordinates $\mathbf{y} \in \mathbb{R}^l$ and the remaining ambient space with $\mathbf{z} \in \mathbb{R}^{d-l}$. To describe an arbitrary bifurcation of interest, we introduce a sufficiently smooth function $g$ and a bifurcation parameter $\epsilon \geq 0$, such that the following system is equivalent to the original ODE:

$$\dot{\mathbf{y}} = \epsilon \mathbf{g}(\mathbf{y}, \mathbf{z}, \epsilon) \qquad \text{(tangent)} \tag{2}$$
$$\dot{\mathbf{z}} = \mathbf{h}(\mathbf{y}, \mathbf{z}, \epsilon) \qquad \text{(normal)} \tag{3}$$

where $\epsilon = 0$ gives the condition for the continuous attractor $\dot{\mathbf{y}} = \mathbf{0}$. We denote the corresponding manifold of $l$ dimensions $\mathcal{M}_0 = \{(\mathbf{y}, \mathbf{z}) \mid \mathbf{h}(\mathbf{y}, \mathbf{z}, 0) = 0\}$.

We need the flow normal to the manifold to be hyperbolic, that is *normally hyperbolic*, meaning that the Jacobians $\nabla_{\mathbf{z}} \mathbf{h}$ evaluated on any point on the $\mathcal{M}_0$ has $d - l$ eigenvalues with their real part uniformly away from zero, and $\nabla_{\mathbf{y}} \mathbf{g}$ has $l$ eigenvalues with zero real parts. More specifically, for continuous attractors, the real part of the eigenvalues of $\nabla_{\mathbf{z}} \mathbf{h}$ will be negative, representing sufficiently strong attractive flow toward the manifold. Equivalently, for the ODE, $\dot{\mathbf{x}} = \mathbf{f}(\mathbf{x})$, the variational system is of constant rank, and has exactly $(d - l)$ eigenvalues with negative real parts and $l$ eigenvalues with zero real parts everywhere along the continuous attractor.

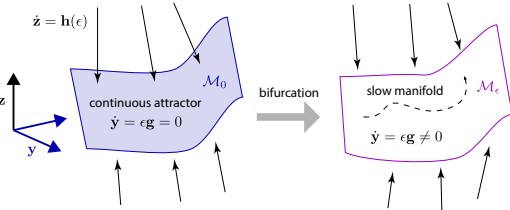

Figure 2: Fenichel's invariant manifold theorem applied to compact continuous attractor guarantees the flow on the slow manifold is locally invariant and continues to be attractive. The dashed line is a trajectory "trapped" in the slow manifold (locally invariant).

When $\epsilon > 0$, the continuous attractor bifurcates away. What can we say about the fate of the perturbed system? The continuous dependence theorem (Chicone, 2006) says that the trajectories will change continuously as a function of $\epsilon$ without a guarantee on how quickly they change. Moreover, the topological structure and the asymptotic behavior of trajectories change discontinuously due to the bifurcation. Surprisingly there is a strong connection in the geometry due to Fenichel's theorem (Fenichel and Moser, 1971). We informally present a special case due to (Jones, 1995):

**Theorem 1** (Fenichel's Invariant Manifold Theorem). *Let $\mathcal{M}_0$ be a connected, compact, normally hyperbolic manifold of equilibria originating from a sufficiently smooth ODE. For a sufficiently small perturbation $\epsilon > 0$, there exists a manifold $\mathcal{M}_\epsilon$ diffeomorphic to $\mathcal{M}_0$ and locally invariant under the flow of (2). Moreover, $\mathcal{M}_\epsilon$ has $\mathcal{O}(\epsilon)$ Hausdoff distance to $\mathcal{M}_0$ and has the same smoothness as $g$ and $h$ in (2).*

The manifold $\mathcal{M}_\epsilon$ is called the *slow manifold* which is no longer necessarily a continuum of equilibria. However, the local invariance implies that trajectories remain within the manifold except potentially at the boundary. Furthermore, the non-zero flow on the slow manifold is slow and given in the $\epsilon \to 0$ limit as $\frac{\mathrm{d}\mathbf{y}}{\mathrm{d}\tau} = \mathbf{g}(c^\epsilon(\mathbf{y}), \mathbf{y}, 0)$ where $\tau = \epsilon t$ is a rescaled time and $c^\epsilon(\cdot)$ parameterizes the $l$ dimensional slow manifold. In addition, the stable manifold of $\mathcal{M}_0$ is similarly approximately maintained (Jones, 1995), allowing the manifold $\mathcal{M}_\epsilon$ to remain attractive.

These conditions are met (up to numerical precision) for the BLA example in Fig. 1B (see Sec. S1.4 for the rerparametrization of the BLA to this form). As a technical note, for the theory apply to a continuous piecewise-linear system, it is required that the invariant manifold is attracting (Simpson, 2018), which is also the case for the BLA. As the theory predicts, BLA bifurcates into a 1-dimensional slow manifold (dark colored regions) that contains fixed points, and overall still attractive. On the contrary, the UBLA does not satisfy the compactness condition, hence the theory does not predict its persistence. Importantly, the "slow" flow on the perturbed system is not bounded.

In practice, the sufficient conditions for RNNs implementing continuous attractors to have this graceful breakdown (like BLA but not UBLA) is for the continuous attractor manifold to be of finite dimension throughout, connected, and bounded. However, in systems with an invariant manifold with dimension at least three, it is possible that a slow manifold with chaotic dynamics is created through a perturbation. This would have as consequence that the perturbed system acquires positive Lyapunov exponents (corresponding to the chaotic orbit), which then can still lead to exploding gradients albeit with very slow flow that has little practical consequence in finite time experiments.

## 2.2 IMPLICATIONS ON MACHINE LEARNING

Extending the memory time constant of RNNs have long been an important area of research with much focus on random weights (Legenstein and Maass, 2007; Goldman, 2009; Toyoizumi and Abbott, 2011; Kerg et al., 2019; Chen et al., 2018; Henaff et al., 2016; Rusch and Mishra, 2021; Arjovsky et al., 2016). Various initializations for the recurrent weights have been proposed to help learning: initialization with the identity matrix (Le et al., 2015), with a random orthogonal matrix (Saxe et al., 2014; Henaff et al., 2016), with a unitary matrix (Arjovsky et al., 2016) and with a block diagonal weight matrix that creates a quasi-periodic system with limit cycles (Sokol et al., 2019). However, despite the capacity to maintain representation of continuous quantities for arbitrary duration of time, continuous attractor mechanism has not been pursued in machine learning research because of its brittleness. The stochasticity in gradients inherited from the training data, regularization strategy, and multi-task learning objectives act as a perturbation on the recurrent dynamics, and continuous attractors break down even if it could be learned. Remedies emerged in machine learning to hard-code continuous-valued memory structures within the RNNs—e.g., the cell state in vanilla LSTM. However, our theory shows that the geometric structure of the manifold and the flow around the manifold play a critical role in enabling gradient descent learning of continuous attractors using standard methods such as backpropagation through time (BPTT) (Toomarian and Barhen, 1991).

It is well known that asymptotic exploding gradients comes from positive Lyapunov exponents (Mikhaeil et al., 2022; Vogt et al., 2022; Engelken et al., 2023). It has also been pointed out that bifurcations can cause arbitrarily large gradients (Doya, 1993) as well as discontinuity in the Lyapunov spectrum (Park et al., 2023). These gradient propagation theories suggested that bifurcations should be avoided, including the continuous attractors.

As far as we know, there is no architecture agnostic theory describing the loss landscape around RNN solutions. We remark that due to the singular nature of the center manifold that supports the continuous attractor, the usual analysis approach of linearization fails. *Our theory successfully connects the invariant manifold theory and the gradient signal propagation theory in RNNs to describe two types of loss landscape around continuous attractor solutions.* In one case, when the theorem holds, the landscape is shallow in all directions due to (asymptotically) vanishing gradients induced by the attractor structure—we have the gracefully degrading continuous attractor. In the other case, we can find examples where the theorem does not hold, and the continuous attractor solution is at the boundary of network configurations with exploding gradients, meaning the loss landscape is very steep in some directions. While exploding gradients would prevent gradient descent to correct for deviations from the optima, for gracefully degrading ones, one can apply restorative forces via gradient descent to be in the vicinity of the brittle continuous attractor solution (see Sec. 3.1.4).

## 2.3 IMPLICATIONS ON NEUROSCIENCE

Continuous attractors are biologically plausible, theoretically elegant, consistent with neural recordings, and avoids the asymptotic exploding and vanishing gradient problem (Park et al., 2023). As a conceptual tool in computational and theoretical neuroscience, continuous attractors are widely used when working memory of continuous values is needed (Dayan and Abbott, 2001; Burak and Fiete, 2009; Khona and Fiete, 2022). When used to accumulate stimulus, continuous attractors are also called neural integrators that are hypothesized to be the underlying computation for the maintenance of eye positions, heading direction, self-location, target location, sensory evidence, working memory, decision variables, to name a few (Seung, 1996; Seung et al., 2000; Romo et al., 1999). Neural representation of continuous values have been observed as persistent activity in the prefrontal cortex of primates, ellipsoid body of the fly, and hypothalamus (Romo et al., 1999; Noorman et al., 2022; Nair et al., 2023). A typical computational implementation of a continuous attractor is a bump attractor network model which requires a mean-field limit (Skaggs et al., 1995; Camperi and Wang, 1998; Renart et al., 2003) and finite sized networks with threshold linear units (Noorman et al., 2022; Spalla et al., 2021), see also Sec. 3.2.

However, the so-called "fine-tuning problem" describing the theoretical and practical brittleness of continuous attractors has long been recognized (Seung, 1998; Park et al., 2023). Since biological neural systems have constantly fluctuating synaptic weights (Shimizu et al., 2021), this has been a big puzzle in the field. There have been efforts and remedies to lessen the degradation for particular implementations, often focusing on keeping the short-term behavior close to the continuous attractor case (Lim and Goldman, 2012; 2013; Boerlin et al., 2013; Koulakov et al., 2002; Renart et al., 2003).

Our theory shows that not all continuous attractors are born equal, and there are gracefully degrading continuous attractors. In finite time, trajectories are well-behaved, contrary to the asymptotic behavior captured by the Lyapunov exponents. Animal behavior is finite time in nature and the longer the temporal distance the harder it is to learn in general. The conditions are favorable in the recurrent neuronal networks: (1) mutual inhibition is widely present and evidence points to inhibition dominated dynamics, (2) the neural state space is bounded due to physiological constraints, namely by a non-negative firing rate below and a maximum firing rate above.

## 3 EXPERIMENTS

To computationally investigate the neighborhood of recurrent dynamical systems that implement continuous attractors, we investigate 5 RNNs that are known a priori to form 1 or 2 dimensional continuous attractors. We consider two topologically distinct temporal integration tasks: (i) linear integration, and (ii) angular integration. For all experiments we used single precision floating point arithmetic and PyTorch.

### 3.1 LINEAR TEMPORAL INTEGRATION TASK

Given a sequence of scalar input, the job of the network is to accumulate the values over time and report the final value at a later time. In the context of perceptual decision-making, subjects can be trained to perform the Poisson clicks task where they have to count the differing number of sensory stimulus events from the left and right side and report the side (Brunton et al., 2013). A linear integrator as a continuous attractor is a natural solution to such a task. We generalize the clicks to have associated continuous-values for the training of RNNs to discourage discrete counting solutions.

We used discrete time representations over $T$ time bins and the stimulus encoded as difference of two non-negative values:

$$I_{t,i} = m_{t,i} \cdot u_{t,i} \qquad t = 1, \ldots, T, \ i = 1, 2 \qquad \text{(continuous clicks)} \qquad (4)$$

$$O_t^* = \sum_{s=0}^{t} (I_{s,1} - I_{s,2}) \qquad t = 1, \ldots, T \qquad \text{(desired output)} \qquad (5)$$

where $m_{t,i}$ are independent Bernoulli random variables with probability $0.2$ and $u_{t,i}$ are independent random variables with uniform distribution on the unit interval. We used mean squared error (MSE) of the 1-dimensional output over time as the loss function over all time bins. We used $T = 100$ time

bins per trial unless specified otherwise. The gradients were computed in batch mode with 1024 randomly generated trials.

### 3.1.1 RNN SOLUTIONS TO THE LINEAR INTEGRATION TASK

We use vanilla RNN implementations with the standard parameterization:

$$\mathbf{x}_t = \sigma(\mathbf{W}_{\text{in}}\mathbf{I}_t + \mathbf{W}\mathbf{x}_{t-1} + \mathbf{b})$$
$$O_t = \mathbf{W}_{\text{out}}\mathbf{x}_t + \mathbf{b}_{\text{out}} \tag{6}$$

where $\mathbf{x}_t \in \mathbb{R}^d$ is the hidden state, $\mathbf{I}_t \in \mathbb{R}^K$ is the input, $\sigma : \mathbb{R} \to \mathbb{R}$ an activation function which acts on each of the hidden dimension, and $\mathbf{W}, \mathbf{b}, \mathbf{W}_{\text{in}}, \mathbf{W}_{\text{out}}, \mathbf{b}_{\text{out}}$ are parameters. Assuming an Euler integration with unit time step, the discrete-time RNN of (6) corresponds to the ODE:

$$\dot{\mathbf{x}} = -\mathbf{x} + \sigma(\mathbf{W}_{\text{in}}\mathbf{I} + \mathbf{W}\mathbf{x} + \mathbf{b}). \tag{7}$$

For tractable analysis, we consider 2 dimensional systems with ReLU activation. We study the three different ReLU RNN implementations of a perfect integrator in a 2 dimensional system, the Identity RNN (iRNN), UBLA and BLA (we refer to the line attractors together as LA). These three networks have same norm in the recurrent matrix $\mathbf{W}$ but not close in the parameter space. On the original clicks task the UBLA and BLA networks count the click differences directly, while iRNN counts the clicks separately and then subtracts these representations through the output mapping. The behaviors of UBLA and BLA in the absence of stimulus are shown in Fig. 1, while the behavior of the iRNN is trivial since there is no flow. These networks are defined as follows.

**Identity RNN (Le et al., 2015)**

$$\mathbf{W}_{\text{in}} = \begin{pmatrix} 1 & 0 \\ 0 & 1 \end{pmatrix}, \ \mathbf{W} = \begin{pmatrix} 1 & 0 \\ 0 & 1 \end{pmatrix}, \ \mathbf{W}_{\text{out}} = \begin{pmatrix} -1 \\ 1 \end{pmatrix}, \ \mathbf{b} = \begin{pmatrix} 0 \\ 0 \end{pmatrix}, \ \mathbf{b}_{\text{out}} = 0. \tag{8}$$

**Unbounded line attractor** We formulate this implementation of a bounded integrator with a parameter that determines step size along line attractor $\alpha$. Together with the parameters for the output bias $\beta$ the parameters determine the capacity of the network. While the line attractor is unbounded from above, it only extends to the center from below. The step size along line attractor $\alpha$ determines the maximum number of clicks as the difference between the two channels; the capacity is $\beta/\alpha$ number of clicks.

$$\mathbf{W}_{\text{in}} = \alpha \begin{pmatrix} -1 & 1 \\ -1 & 1 \end{pmatrix}, \ \mathbf{W} = \begin{pmatrix} 0 & 1 \\ 1 & 0 \end{pmatrix}, \ \mathbf{W}_{\text{out}} = \frac{1}{2\alpha} \begin{pmatrix} 1 \\ 1 \end{pmatrix}, \ \mathbf{b} = \begin{pmatrix} 0 \\ 0 \end{pmatrix}, \ \mathbf{b}_{\text{out}} = -\frac{\beta}{\alpha}. \tag{9}$$

**Bounded line attractor** Similarly as for UBLA, the BLA has a parameter that determines step size along line attractor $\alpha$. Analogously as for UBLA, these parameters determine the capacity of the network. The inputs push the input along the line attractor in two opposite directions, see below. UBLA and BLA need to be initialized at $\beta(1,1)$ and $\frac{\beta}{2}(1,1)$, respectively, for correct decoding, i.e., output projection.

$$\mathbf{W}_{\text{in}} = \alpha \begin{pmatrix} -1 & 1 \\ 1 & -1 \end{pmatrix}, \ \mathbf{W} = \begin{pmatrix} 0 & -1 \\ -1 & 0 \end{pmatrix}, \ \mathbf{W}_{\text{out}} = \frac{1}{2\alpha} \begin{pmatrix} 1 \\ -1 \end{pmatrix}, \ \mathbf{b} = \beta \begin{pmatrix} 1 \\ 1 \end{pmatrix}, \ \mathbf{b}_{\text{out}} = 0. \tag{10}$$

### 3.1.2 ASYMMETRIC LOSS LANDSCAPE REFLECTING DYNAMICS AFTER BIFURCATION

To illustrate the effect of bifurcations from the continuous attractor solution, we take a 1-dimensional slice of the loss surface, see Fig. 3B. Specifically, we continuously vary one of the entries of the self-recurrent connection matrix: $\mathbf{W}_{1,1} \leftarrow \mathbf{W}_{1,1} + \Delta$. At any $\Delta \neq 0$, the continuous attractor disappears and the spontaneous dynamics of the networks show convergent and/or divergent behavior at exponential rates. Therefore, as the number of time steps in a trial increases, the error in the output also exponentially converge or diverge in a corresponding manner. As can be seen in Fig. 3B, for UBLA and iRNN, $\Delta > 0$ perturbations shows exponentially increasing loss and corresponds to an exploding gradient dynamical system. In all other cases, including all perturbations of BLA, leads to vanishing gradient, hence the loss is bounded. Note also the high curvature of the loss landscape around the optimal solution indicating that the slow manifold may only be maintained in a small neighborhood around the optimal solution, especially for the LAs.

### 3.1.3 BIFURCATION PROBABILITY AND RANDOM PERTURBATIONS OF BLA

We consider all parametrized perturbations of the form $\mathbf{W} \leftarrow \mathbf{W} + \mathbf{V}$ for a random matrix $\mathbf{V} \in \mathbb{R}^{2 \times 2}$ to the BLA. The BLA can bifurcate in the following systems, characterized by their invariant sets: a system with single stable fixed point, a system with three fixed points (one unstable and two stable) and a system with two fixed points (one stable and the other a half-stable node) and a system with a (rotated) line attractor. Only the first two bifurcations (Fig. 1) can happen with nonzero chance for the type of random perturbations we consider. The perturbations that leave the line attractor intact or to lead to a system with two fixed points have measure zero in the parameter space. The perturbation that results in one fixed point happen with probability $\frac{3}{4}$, while perturbations lead to a system with three fixed points with probability $\frac{1}{4}$, see Sec. S1.2. The (local) invariant manifold manifold is indeed persistent for the BLA and homeomorphic to the original (the bounded line).

### 3.1.4 MAINTAINING A NEURAL INTEGRATOR

The theory of persistent invariant manifolds for compact continuous attractors suggests that the BLA should have bounded gradients (unlike UBLA and iRNN) and hence it should be easier to maintain it in the presence of noise. To investigate the differential effect of stochastic gradient descent (SGD) on the three neural integrator models, we performed three learning experiments using the continuous-valued click integration task. The input and output are defined as in Eqs. 4 and 5 with $I_{t,i} = 0$ for $t = 11, \ldots, T$. We investigate the effects of perturbations of the recurrent matrix on the learning of the parameters during gradient descent starting from the perfect solutions to the task. Gradient step were taken with a fixed gradient step size $\lambda$ (learning rate). We set $\alpha = 1$ and $\beta = 20$ in Eq 9 and 10. The hidden state at the start of a trial is a learnable parameter.

In the first experiment, Gaussian random noise is injected to all parameters inducing a constant diffusion of the parameters, which emulates the biological synaptic variability. This type of noise is directly applied to the weights as $\mathbf{W} \leftarrow \mathbf{W} + \mathbf{V}$ with $\mathbf{V}_{i,j} \sim \mathcal{N}(0, \sigma)$. To dissociate the effect of misadjustment from gradient descent and external perturbation, we measured the effect of a single perturbation on the learning dynamics. Fig. 3C shows that for all networks, gradient descent (with constant learning rate, chosen from a grid search) was able to counter the diffusion. BLA and UBLA with learning have superior misadjustment compared to iRNN and compared to perturbations without learning, while the BLA has the broadest range of learning rates that are optimal and far away from exploding gradients (Fig. 3A). BLA has a slight advantage in terms of a smaller spread of MSE compared to UBLA. The invariant manifold of the BLA is persistent throughout learning in many cases, see S8 and Fig. S12. However, the gradients are not pointing towards the BLA but to one of the bifurcations of the BLA (see Supp Fig. S10, Fig. S13 and Supp. Fig. S14). We determine the alignment of the gradient as the cosine similarity of the gradient step with the vector in recurrent parameter space that points towards the initial parameters at every gradient step and use a cutoff of a maximum deviation of $45°$ as aligned gradients with the optimal solution direction. iRNN often finds a different optimum (it settles at a part of the state space that is at a non-zero distance from the initial recurrent matrix and bias (Fig. 3D and Fig S12). UBLA can stay close to the initial solution for a small enough learning rate (Fig. 3D and E) and maintains a slower flow than the BLA (Fig. S15).

We calculated the loss on a batch of inputs for various noise levels $\sigma$ for all three noise types (Fig. S6). We chose a matched noise level per integrator that corresponded to a set average loss averaged over 200 weight perturbations (see also in Sec. S3). This way of matching noise level to induce the same loss should be a universal approach to be able to compare the performance of different networks.

For the matched noise level, we find the optimal learning rate for each network separately. The optimal learning rates for the input-type noise experiments were chosen from a set of values ($\{(1 + j\frac{1}{4}))10^{-i}\}_{i=4,\ldots 10, j=1,2,3}$)) based on best performance of the task, measured as mean MSE of the last ten epochs averaged over ten runs. The slow manifold that is created after perturbations provides gradients that can counteract parameter diffusions for all networks (on short trials), even for the ones that have the potential for exploding gradients (Fig. 3A and C). We use the normed difference to the initial parameters at every gradient step as proxy for the misadjustment from the optimal solution (Fig. 3D and E). We further show that all networks converge to a different (from the initialization), non-optimal, solution as they settle in a regime in parameter space that has a higher norm difference with the initial parameters of the neural integrator in ten different runs with the same random seed

for the noise for the three integrators (Fig. 3D and E). We conclude therefore that, in practice, it is difficult to maintain any of the continuous attractors.

Note that exploding gradients can be seen for UBLA manifested as the bimodal distribution of the gradients in Fig. 3F. This does lead to faster divergence (for lower learning rate) but has on the other hand the benefit of providing useful gradients to maintain the (local) solution around the optimal solution, which explains the superior performance at the optimal learning rate for the UBLA (Fig. 3A), on this timescale for the trial that we investigated. Also in the presence of input and internal noise the UBLA has a higher tendency to have exploding gradients for lower learning rates, see Fig. S9. We hypothesise that the negative effect of exploding gradients shows only for longer trials.

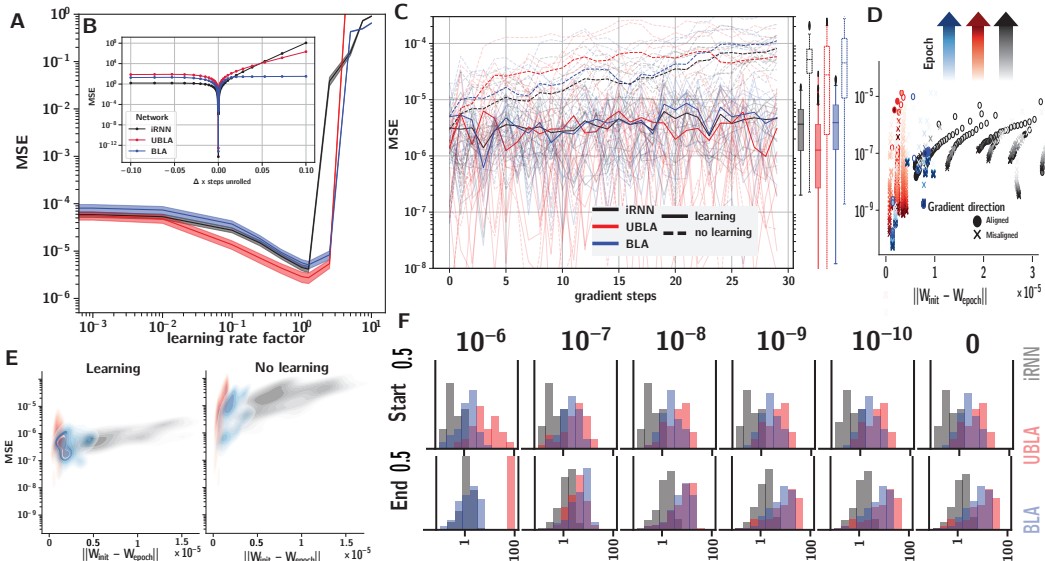

Figure 3: Comparing three continuous attractor solutions on the click integration task of $T = 100$ time steps. (A) MSE distribution during learning with different learning rates. (B) Loss landscape is steeper around the attractors. The BLA has a bounded loss in its neighborhood. (C) MSE distribution during learning and in the presence of noise. Learning counteracts diffusion in all three-types of initializations. (D) All networks converge to local non-optimal solutions after a single perturbation in 30 gradient steps. (E) Distance of parameters to original during learning with noise (left) and without learning (right). (F) Gradient distribution at the beginning (upper) and end (lower) of trials.

## 3.2 ANGULAR INTEGRATION NETWORKS

For circular variables such as the goal-direction or head-direction needed for spatial navigation in 2D, the temporal integration and working memory functions are naturally solved by a ring attractor (continuous attractor with a ring topology). Continuous attractor models of the head-direction representation suggest that the representation emerges from the interactions of recurrent connections among neurons that form a ring-like structure (Zhang, 1996; Noorman et al., 2022; Ajabi et al., 2023). Since continuous attractors are susceptible to noise and perturbations the precise representation of the head direction can in principle be disrupted easily. We demonstrate the consequences of the Persistence Theorem in two models with a continuous ring attractor.

The first model we analyzed is a simple (non-biological) system that has a ring attractor we analysed is defined by the following ODE: $\dot{r} = r(1 - r)$, $\dot{\theta} = 0$. This system has as fixed points the origin and the ring with radius one centered around zero, i.e., $(0, 0) \cup \{(1, \theta) \mid \theta \in [0, 2\pi)\}$. We investigate bifurcations caused by parametric and bump perturbations of the ring invariant manifold (see Sec. S9), which is bounded and boundaryless. All perturbations maintain the invariant manifold (Fig. 4B).

Second, we investigated perturbations of a continuous ring attractor proposed as a model for the head direction representation in fruitflies (Noorman et al., 2022). As this continuous ring attractor is bounded its invariant manifold persists and, hence, no divergent orbits are created under small

perturbations to this system. Furthermore, because the ring attractor is boundaryless it is both forward and backward invariant, i.e. hence it is invariant and trajectories never leave the persistent invariant manifold (Wiggins, 1994). This model is composed of $N$ heading-tuned neurons with preferred headings $\theta_j \in \{\frac{2\pi i}{N}\}_{i=1\dots N}$ radians (see Supp. Sec S10). For sufficiently strong local excitation (given by the parameter $J_E$) and broad inhibition ($J_I$), this network will generate a stable bump of activity, one corresponding to each head direction. This continuum of fixed points forms a one dimensional manifold homeomorphic to the circle.

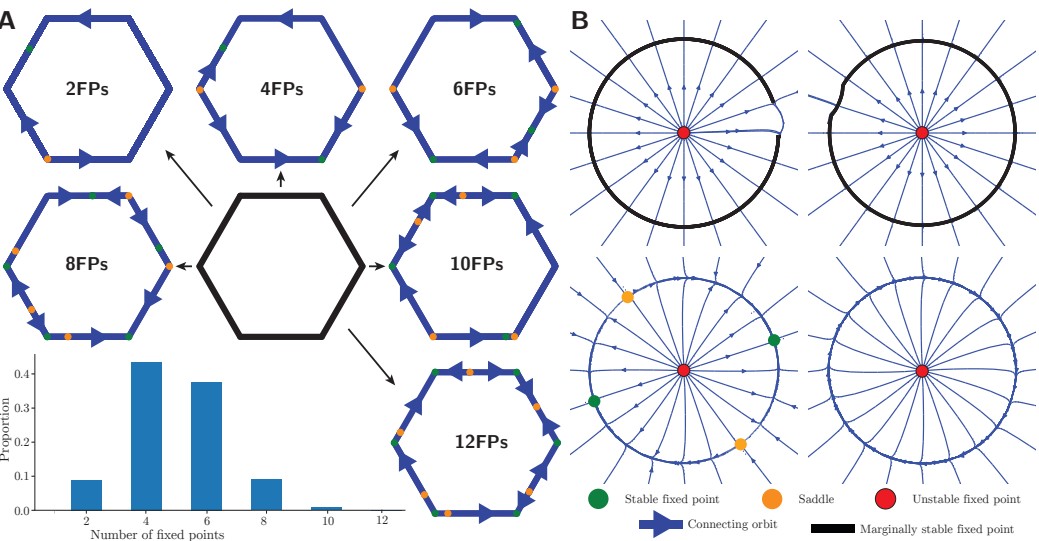

Figure 4: Characterization of bifurcations of two ring attractors. (A) Perturbations to the ring attractor (Noorman et al., 2022). The ring attractor can be perturbed in systems with an even number of fixed points (FPs) up to $2N$ (stable and saddle points are paired). (B) Perturbations to a simple implementation of a ring attractor lead to bifurcations that all leave the invariant manifold intact.

We evaluate the effect of parametric perturbations of the form $\mathbf{W} \leftarrow \mathbf{W} + \mathbf{V}$ with $\mathbf{V}_{i,j} \overset{iid}{\sim} \mathcal{N}(0, \frac{1}{100})$ on a network of size $N = 6$ with $J_E^* = 4$ and $J_I = -2.4$ by identifying all bifurcations (Sec. S9). We found that the ring (consisting of infinite fixed points) can be perturbed into systems with between 2 and 12 fixed points (Fig. 4A). As far as we know, this bifurcation from a ring of equilibria to a saddle and node has not been described previously in the literature. The probability of each type of bifurcation was numerically estimated. There are several additional co-dim 1 bifurcations with measure zero (see Fig. S16).

## 4 DISCUSSION

The attractive manifold of equilibria in continuous attractor networks provides two key functions continuous memory and propagation of gradient through time. Although the corresponding configurations are measure zero, we showed that the when the invariant manifold theorem holds, the finite time behavior of the trajectories and the gradient through time only slowly breakdown. We investigated the neighborhood of the continuous attractor networks and analyzed diverse bifurcations in 5 example systems. There were surprisingly diverse bifurcations which provide additional insights to vanishing and exploding gradient regimes that the network visits in the presence of stochasticity in synapses and learning signals. In particular, we showed that some RNNs are devoid of bifurcations that lead to exploding gradients with non-zero measure.

As the theory predicts, our numerical experiments demonstrate the properties of loss landscape and gradient near the fine-tuned system. However, after small perturbations, plain gradient descent typically converges to a non-continuous attractor solution, indicating that the homeostatic restoration of continuous attractor may be challenging. Based on our observations, we cannot conclude that continuous attractors solutions are universally brittle. Further research on finding continuous attractor networks that may allow restorative learning for larger perturbations in the parameter space is needed.

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

## SUPPLEMENTARY MATERIAL

## S1    BIFURCATION ANALYSIS OF THE LINE ATTRACTORS

### S1.1    UNBOUNDED LINE ATTRACTOR

**Stabilty of the fixed point with full support**    We investigate how perturbations to the bounded line affect the Lyapunov spectrum. We calculate the eigenspectrum of the Jacobian:

$$\det[W' - (1 + \lambda)\mathbb{I}] = (\epsilon_{11} - 1 - \lambda)(\epsilon_{22} - 1 - \lambda) - (\epsilon_{12} + 1)(\epsilon_{21} + 1)$$
$$= \lambda^2 - (2 + \epsilon_{11} + \epsilon_{22})\lambda - \epsilon_{11} - \epsilon_{22} + \epsilon_{11}\epsilon_{22} - \epsilon_{12} - \epsilon_{21} - \epsilon_{12}\epsilon_{21}$$

Let $u = -(2 + \epsilon_{11} + \epsilon_{22})$ and $v = -\epsilon_{11} - \epsilon_{22} + \epsilon_{11}\epsilon_{22} - \epsilon_{12} - \epsilon_{21} - \epsilon_{12}\epsilon_{21}$

There are only two types of invariant set for the perturbations of the line attractor. Both have as invariant set a fixed point at the origin. What distinguishes them is that one type of perturbations leads to this fixed point being stable while the other one makes it unstable.

### S1.2    BOUNDED LINE ATTRACTOR

**Input**    Parameter that determines step size along line attractor $\alpha$. The size determines the maximum number of clicks as the difference between the two channels. This pushes the input along the line "attractor" in two opposite directions, see below.

**Stability of the fixed points**    We perform the stability analysis for the part of the state space where $Wx > 0$. There, the Jacobian is

$$J = -\begin{pmatrix} 1 & 1 \\ 1 & 1 \end{pmatrix} \tag{11}$$

We apply the perturbation

$$W' = \begin{pmatrix} 0 & -1 \\ -1 & 0 \end{pmatrix} + \epsilon \tag{12}$$

with

$$\epsilon = \begin{pmatrix} \epsilon_{11} & \epsilon_{12} \\ \epsilon_{21} & \epsilon_{22} \end{pmatrix} \tag{13}$$

The eigenvalues are computed as

$$\det[W' - (1 + \lambda)\mathbb{I}] = (\epsilon_{11} - 1 - \lambda)(\epsilon_{22} - 1 - \lambda) - (\epsilon_{12} - 1)(\epsilon_{21} - 1)$$
$$= \lambda^2 + (2 - \epsilon_{11} - \epsilon_{22})\lambda - \epsilon_{11} - \epsilon_{22} + \epsilon_{11}\epsilon_{22} + \epsilon_{12} + \epsilon_{21} - \epsilon_{12}\epsilon_{21}$$

Let $u = 2 - \epsilon_{11} - \epsilon_{22}$ and $v = -\epsilon_{11} - \epsilon_{22} + \epsilon_{11}\epsilon_{22} + \epsilon_{12} + \epsilon_{21} - \epsilon_{12}\epsilon_{21}$

$$\lambda = \frac{-u \pm \sqrt{u^2 - 4v}}{2} \tag{14}$$

Case 1: $\text{Re}(\sqrt{u^2 - 4v}) < -u$, then $\lambda_{1,2} < 0$

Case 2: $\text{Re}(\sqrt{u^2 - 4v}) > -u$, then $\lambda_1 < 0$ and $\lambda_2 > 0$

Case 3: $v = 0$, then $\lambda = \frac{1}{2}(-u \pm u)$, i.e., $\lambda_1 = 0$ and $\lambda_2 = -u$

$$\epsilon_{11} = -\epsilon_{22} + \epsilon_{11}\epsilon_{22} + \epsilon_{12} + \epsilon_{21} - \epsilon_{12}\epsilon_{21} \tag{15}$$

We give some examples of the different types of perturbations to the bounded line attractor. The first type is when the invariant set is composed of a single fixed point, for example for the perturbation:

$$\epsilon = \frac{1}{10} \begin{pmatrix} -2 & 1 \\ 1 & -2 \end{pmatrix} \tag{16}$$

The second type is when the invariant set is composed of three fixed points:

$$\epsilon = \frac{1}{10} \begin{pmatrix} 1 & -2 \\ -2 & 1 \end{pmatrix} \tag{17}$$

The third type is when the invariant set is composed of two fixed points, both with partial support.

$$b' = \frac{1}{10} \begin{pmatrix} 1 & -1 \end{pmatrix} \tag{18}$$

The fourth and final type is when the line attractor is maintained but rotated:

$$\epsilon = \frac{1}{20} \begin{pmatrix} 1 & 10 \\ 10 & 1 \end{pmatrix} \tag{19}$$

**Theorem 2.** *All perturbations of the bounded line attractor are of the types as listed above.*

*Proof.* We enumerate all possibilities for the dynamics of a ReLU activation network with two units. First of all, note that there can be no limit cycle or chaotic orbits.

Now, we look at the different possible systems with fixed points. There can be at most three fixed points (**?**, Corollary 5.3). There has to be at least one fixed point, because the bias is non-zero.

General form (example):

$$\epsilon = \frac{1}{10} \begin{pmatrix} -2 & 1 \\ 1 & -2 \end{pmatrix} \tag{20}$$

One fixed point with full support:

In this case we can assume $W$ to be full rank.

$$\dot{x} = \mathrm{ReLU}\left[ \begin{pmatrix} \epsilon_{11} & \epsilon_{12} \\ \epsilon_{21} & \epsilon_{22} \end{pmatrix} \begin{pmatrix} x_1 \\ x_2 \end{pmatrix} + \begin{pmatrix} 1 \\ 1 \end{pmatrix} \right] - \begin{pmatrix} x_1 \\ x_2 \end{pmatrix} = 0$$

Note that $x > 0$ iff $z_1 := \epsilon_{11} x_1 + (\epsilon_{12} - 1) x_2 - 1 > 0$. Similarly for $x_2 > 0$.

So for a fixed point with full support, we have

$$\begin{pmatrix} x_1 \\ x_2 \end{pmatrix} = A^{-1} \begin{pmatrix} -1 \\ -1 \end{pmatrix} \tag{21}$$

with

$$A := \begin{pmatrix} \epsilon_{11} - 1 & \epsilon_{12} - 1 \\ \epsilon_{21} - 1 & \epsilon_{22} - 1 \end{pmatrix}.$$

Note that it is not possible that $x_1 = 0 = x_2$.

Now define

$$B := A^{-1} = \frac{1}{\det A} \begin{pmatrix} \epsilon_{22} - 1 & 1 - \epsilon_{12} \\ 1 - \epsilon_{21} & \epsilon_{11} - 1 \end{pmatrix}$$

with

$$\det A = \epsilon_{11}\epsilon_{22} - \epsilon_{11} - \epsilon_{22} - \epsilon_{12}\epsilon_{21} + \epsilon_{12} + \epsilon_{21}.$$

Hence, we have that $x_1, x_2 > 0$ if $B_{11} + B_{12} > 0$, $B_{21} + B_{22} > 0$ and $\det A > 0$ and $B_{11} + B_{12} < 0$, $B_{21} + B_{22} < 0$ and $\det A < 0$.

This can be satisfied in two ways, If $\det A > 0$, this is satisfied if $\epsilon_{22} > \epsilon_{12}$ and $\epsilon_{11} > \epsilon_{21}$, while if $\det A > 0$, this is satisfied if $\epsilon_{22} < \epsilon_{12}$ and $\epsilon_{11} < \epsilon_{21}$. This gives condition 1.

Finally, we investigate the condition that specify that there are fixed points with partial support. If $x_1 = 0$ then $(\epsilon_{22} - 1) x_2 + 1 = 0$ and $z_1 < 0$. From the equality, we get that $x_2 = \frac{1}{1-\epsilon_{22}}$. From the inequality, we get $(\epsilon_{12} - 1) x_2 + 1 \geq 0$, i.e. $\frac{1}{1-\epsilon_{12}} \geq x_2$. Hence,

$$\frac{1}{1 - \epsilon_{12}} \geq \frac{1}{1 - \epsilon_{22}}$$

and thus

$$\epsilon_{22} \leq \epsilon_{12}. \tag{22}$$

Similarly to have a fixed point $x^*$ such that $x_2^* = 0$, we must have that

$$\epsilon_{11} \leq \epsilon_{21}. \tag{23}$$

Equation 22 and 23 together form condition 2.

Then, we get the following conditions for the different types of bifurcations:

1. If condition 1 is violated, but condition 2 is satisfied with exactly oner strict inequality, there are two fixed points on the boundary of the admissible quadrant.

2. If condition 1 is violated, and only one of the subconditions of condition 2 is satisfied, there is a single fixed point on one of the axes.

3. If condition 2 is violated, there is a single fixed point with full support.

4. If both conditions are satisfied, there are three fixed points.

We now look at the possibility of the line attractor being preserved. This is the case if $v = 0$. It is not possible to have a line attractor with a fixed point off of it for as there cannot be disjoint fixed points that are linearly dependent (Morrison et al., Lemma 5.2). □

## S1.3 STRUCTURE OF THE PARAMETER SPACE

Table 1: Summary of the conditions for the different bifurcations.

|    | 1FP (full) | 1FP (partial) | 3FPs | 2FPs | LA |
|----|------------|---------------|------|------|----|
| C1 | ✓ | ✗ | ✓ | ✗ | ✗ |
| C2 | ✗ | only Eq22 or 23 | ✓ | ✓ | ✗ |

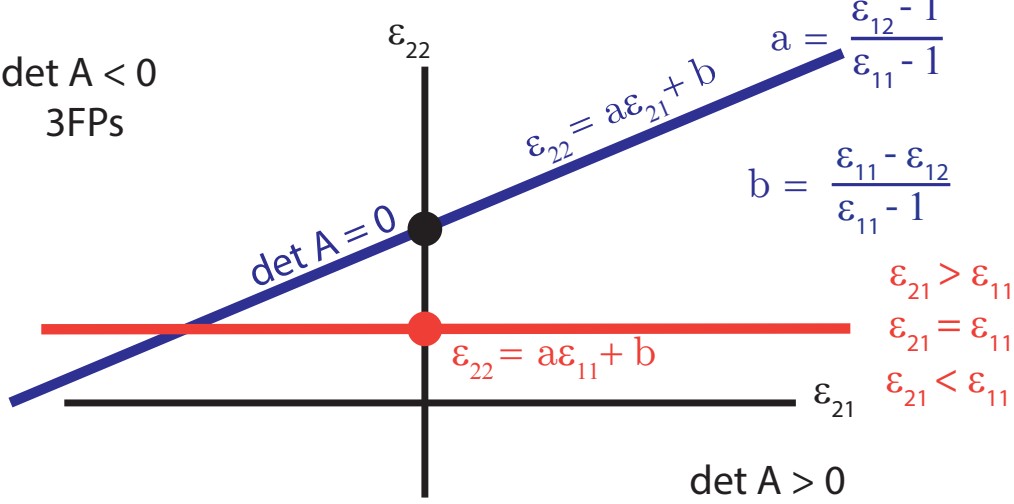

Figure S5: A slice of the parameter space of the BLA for a fixed $\epsilon_{11}$ and $\epsilon_{12}$.

### S1.3.1 PROBABILITY OF BIFURCATION TYPES

We check what proportion of the bifurcation parameter space is constituted with bifurcations of the type that result in three fixed points.

The conditions are

$$0 < \epsilon_{11}\epsilon_{22} - \epsilon_{11} - \epsilon_{22} - \epsilon_{12}\epsilon_{21} - \epsilon_{12} - \epsilon_{21},$$
$$\epsilon_{22} \le \epsilon_{12},$$
$$\epsilon_{11} \le \epsilon_{21}.$$

Because

$$\epsilon_{22} \le \epsilon_{12},$$
$$\epsilon_{11} \le \epsilon_{21}.$$

we always have that

$$0 < \epsilon_{11}\epsilon_{22} - \epsilon_{11} - \epsilon_{22} - \epsilon_{12}\epsilon_{21} - \epsilon_{12} - \epsilon_{21}.$$

This implies that this bifurcation happens with $\frac{1}{4}$ probability in a $\epsilon$-ball around the BLA neural integrator with $\epsilon < 1$.

### S1.4  FAST-SLOW FORM

We transform the state space so that the line attractor aligns with the $y$-axis. So, we apply the affine transformation $R_\theta(x - \frac{1}{2})$ with the rotation matrix $R_\theta = \begin{bmatrix} \cos\theta & -\sin\theta \\ \sin\theta & \cos\theta \end{bmatrix} = \frac{1}{\sqrt{2}} \begin{bmatrix} 1 & 1 \\ -1 & 1 \end{bmatrix}$ where we have set $\theta = -\frac{\pi}{4}$. So we perform the transformation $x \to x' = R_\theta(x - \frac{1}{2})$ and so we have $x = R_\theta^{-1} x' + \frac{1}{2}$ with $R_\theta^{-1} = R_{-\theta}$. Then we get that

$$R_\theta^{-1}\dot{x}' = \text{ReLU}\left(W(R_\theta^{-1}x' + \frac{1}{2}) + 1\right) - R_\theta^{-1}x' - \frac{1}{2}. \tag{24}$$

For a perturbed connection matrix $W = \begin{bmatrix} \epsilon & -1 \\ -1 & 0 \end{bmatrix}$ we get

$$R_\theta^{-1}\dot{x}' = \text{ReLU}\left(\frac{1}{\sqrt{2}}\begin{bmatrix} \epsilon & -1 \\ -1 & 0 \end{bmatrix}\left(\begin{bmatrix} 1 & -1 \\ 1 & 1 \end{bmatrix}x' + \frac{1}{2}\right) + 1\right) - \frac{1}{\sqrt{2}}\begin{bmatrix} 1 & -1 \\ 1 & 1 \end{bmatrix}x' - \frac{1}{2} \tag{25}$$

$$\dot{x}' = \begin{bmatrix} -1 & 1 \\ 1 & 1 \end{bmatrix}\left(\frac{1}{2}\begin{bmatrix} \epsilon - 1 & -\epsilon - 1 \\ -1 & 1 \end{bmatrix}x' + \frac{1}{2\sqrt{2}}\begin{bmatrix} \epsilon - 1 \\ -1 \end{bmatrix} + \begin{bmatrix} 1 \\ 1 \end{bmatrix} - \frac{1}{2}\begin{bmatrix} 1 \\ 1 \end{bmatrix}\right) - x' \tag{26}$$

$$\dot{x}' = \left(\begin{bmatrix} -2 & 0 \\ 0 & 0 \end{bmatrix} + \frac{\epsilon}{2}\begin{bmatrix} 1 & -1 \\ -1 & 1 \end{bmatrix}\right)x' + \frac{1}{2\sqrt{2}}\begin{bmatrix} \epsilon \\ -\epsilon \end{bmatrix} \tag{27}$$

## S2  SMOOTHER ACTIVATION FUNCTIONS

It is well-known that activation functions ($\sigma$ in Eqs. 6 and 7), which can take many forms, play a critical role in propagating gradients effectively through the network and backwards in time (Jagtap and Karniadakis, 2023; Ramachandran et al., 2017; Hayou et al., 2019). Activation functions that are $C^r$ for $r \ge 1$ are the ones to which the Persistence Theorem applies. The Persistence Theorem further specifies how the smoothness of the activation can have implications on the smoothness of the persistent invariant manifold. For situations where smoothness of the persistent invariant manifold is of importance, smoother activation functions might be preferable, such as the Exponential Linear Unit (ELU)(Clevert et al., 2015) or the Continuously Differentiable Exponential Linear Units (CELU) (Barron, 2017).

# S3 INTERNAL AND INPUT NOISE AND NOISE LEVEL MATCHING

We investigate the effects of two other types of noise on the learning of the parameters during gradient descent starting from the perfect solutions to the task. We investigated the effect of learning when perturbations are induced by the backpropagated gradient which is structured by the recurrent dynamics in the following two ways. The second type of noise is injected into the input $x_{i,t} + \epsilon_{i,t}$ with $\epsilon \sim \mathcal{N}(0, \sigma)$. To inject noisy gradients naturally, we added noise to the input to the first 10 time steps during the trial that were not integrated in the target output $O_t^*$ (Eq. 5). The third type of noise is injected into the hidden state $h_{i,t} + \epsilon_{i,t}$ with $\epsilon \sim \mathcal{N}(0, \sigma)$ for $t = 1, \ldots, T$ and $i = 1, 2$.

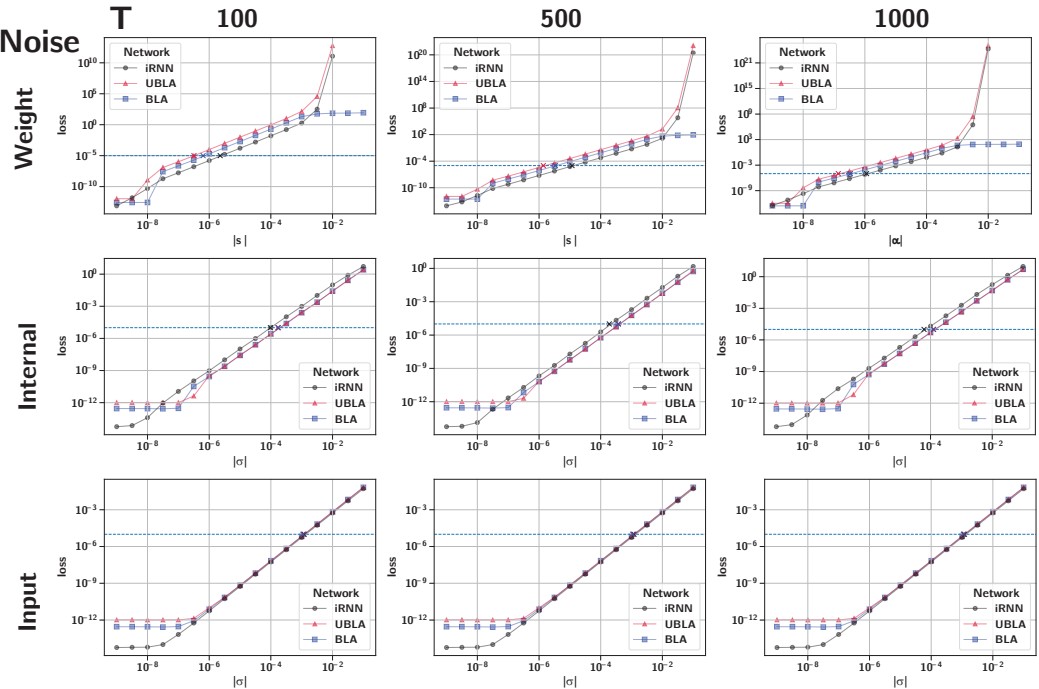

Figure S6: For various values the loss was calculated for the three types of noise. The matched noise levels were chosen based on these curves.

## S4 CONTINUOUS ATTRACTOR SOLUTIONS IN CLICK INTEGRATION TASKS WITH NOISE IN THE WEIGHTS

For the SGD, the last output of the network after $T$ steps was taken to calculate the loss based on the mean squared error (MSE) over a batch of 1024 trials.

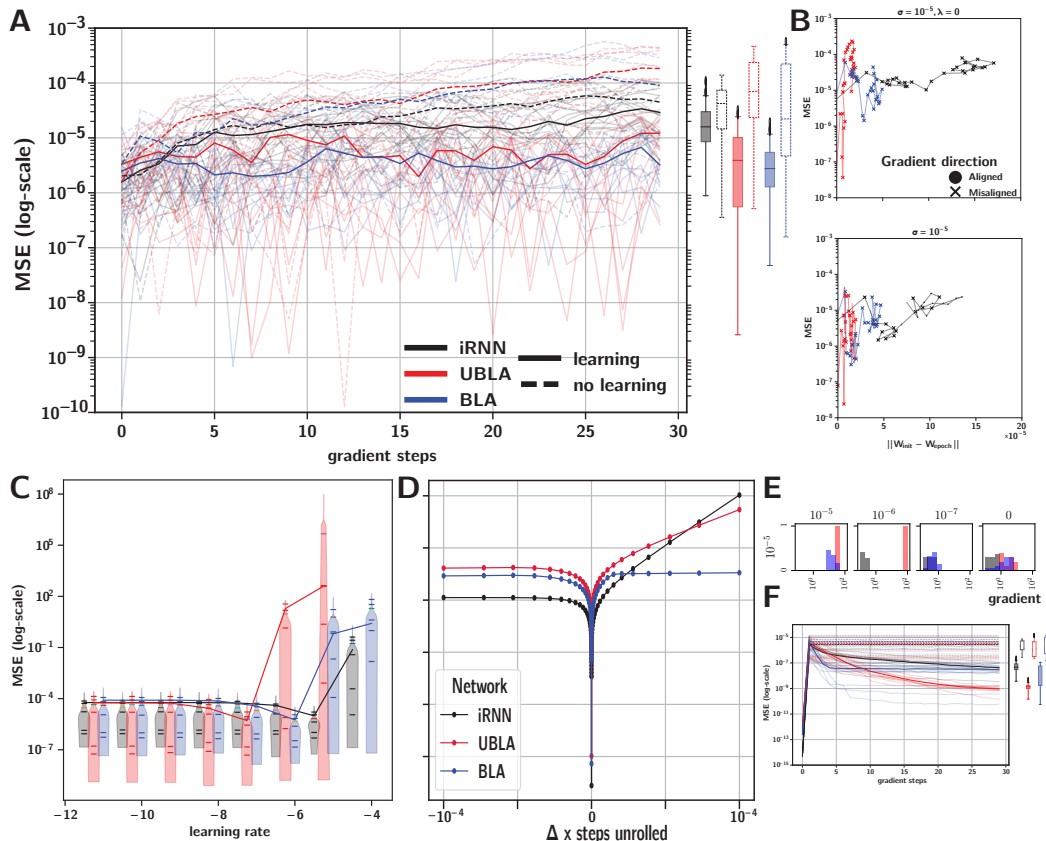

Figure S7: Comparing three continuous attractor solutions to the click integration task for a length of $T = 100$ time steps. (A) Effect of gradient descent in repairing the continuous attractor. RNNs without gradient descent (dashed line) are shown for reference. Box plots show distribution of the loss for the last 10 steps. Averages (thick lines) over 10 simulations (thin lines) are shown for each network. (B) Changes to the recurrent parameters (matrix and bias), without (upper) and with (lower) learning (with the optimal learning rates). iRNN converges to a different solution. (C) The distribution of the MSE for different learning rates. The dip in the MSE defines the optimal learning rate for each of the three neural integrators. (D) Single parameter perturbation showing exploding gradients for iRNN and UBLA. (E) Distribution of gradients shows bimodal distribution for UBLA. (F) Interleaved weight perturbations showing quick recovery for BLA and and slow for iRNN and UBLA.

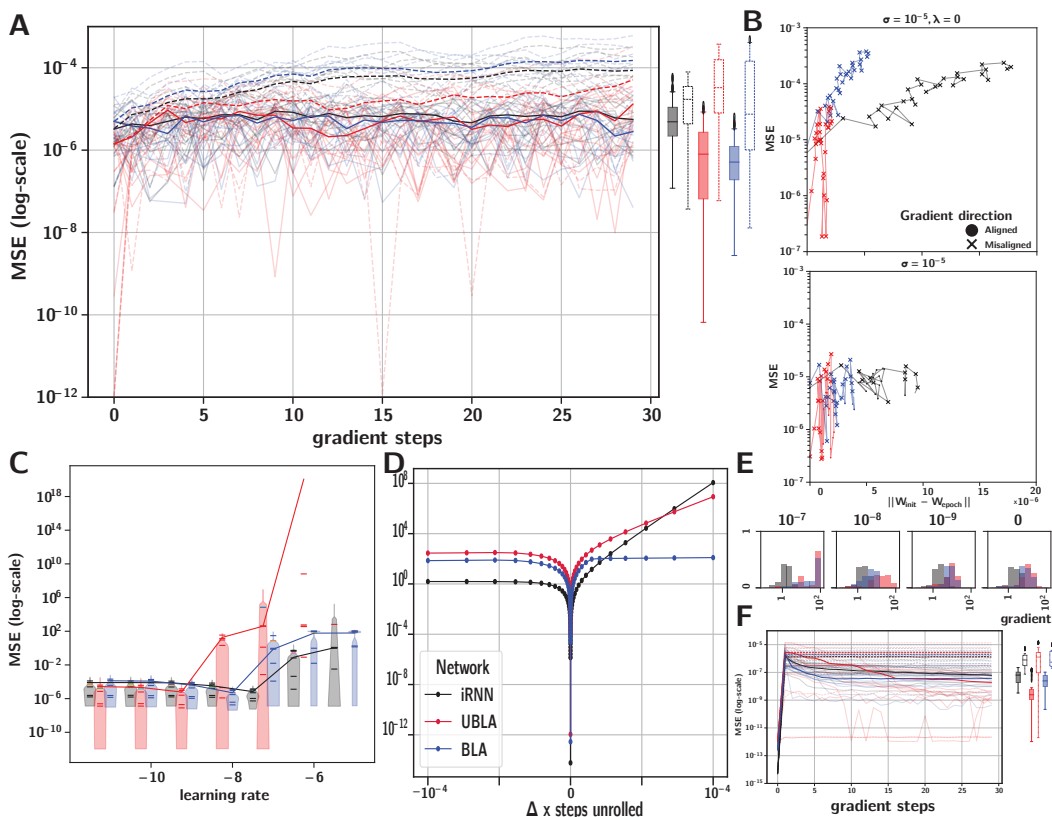

Figure S8: Comparing three continuous attractor solutions to the click integration task for a length of $T = 1000$ time steps. (A) Effect of gradient descent in repairing the continuous attractor. RNNs without gradient descent (dashed line) are shown for reference. Box plots show distribution of the loss for the last 10 steps. (B) Changes to the recurrent parameters (matrix and bias), without (upper) and with (lower) learning (with the optimal learning rates). iRNN converges to a different solution. (C) The distribution of the MSE for different learning rates. The dip in the MSE defines the optimal learning rate for each of the three neural integrators. (D) Single parameter perturbation showing exploding gradients for iRNN and UBLA. (E) Distribution of gradients shows bimodal distribution for UBLA. (F) Interleaved weight perturbations showing quick recovery for BLA and and slow for iRNN and UBLA.

## S5   STABILITY OF THE NEURAL INTEGRATORS FOR DIFFERENT LEARNING RATES

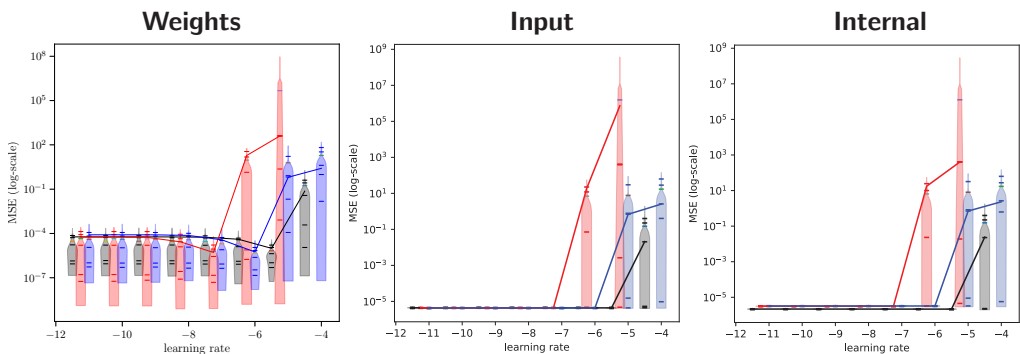

Figure S9: Distribution of MSE for the three noisy types for different learning rates.

## S6 TRAJECTORIES OF THE NEURAL INTEGRATORS IN THE RECURRENT NETWORK SPACE

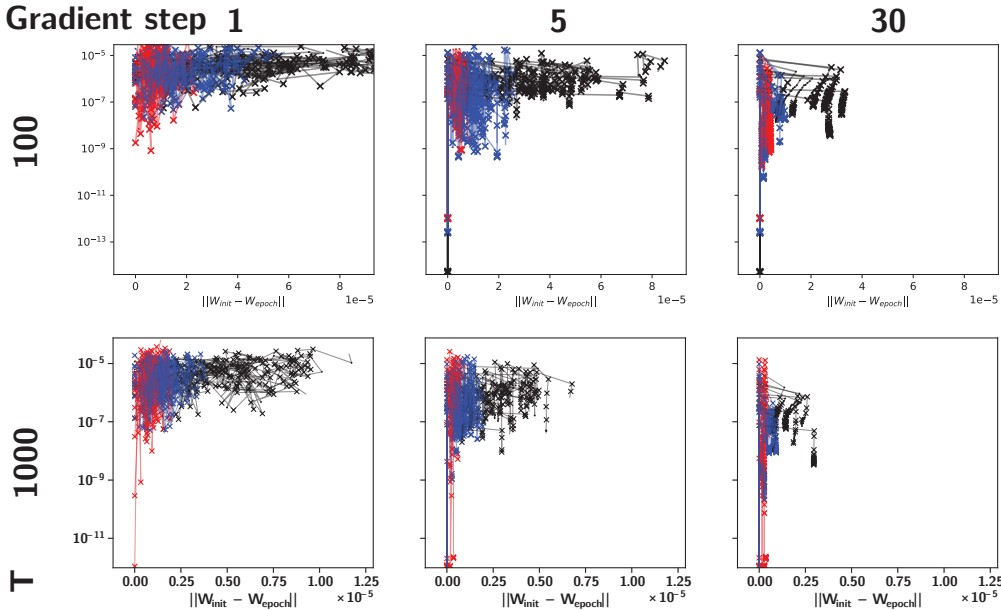

Figure S10: Trajectories of learning in parameter space relative to the initial recurrent parameters. The LAs follow a trajectory that is orthogonal to the initial parameters, but that yet decreases the MSE.

## S7    INFLUENCE OF THE DIFFERENT NOISE TYPES ON THE FOUND SOLUTIONS FOR THE NEURAL INTEGRATORS

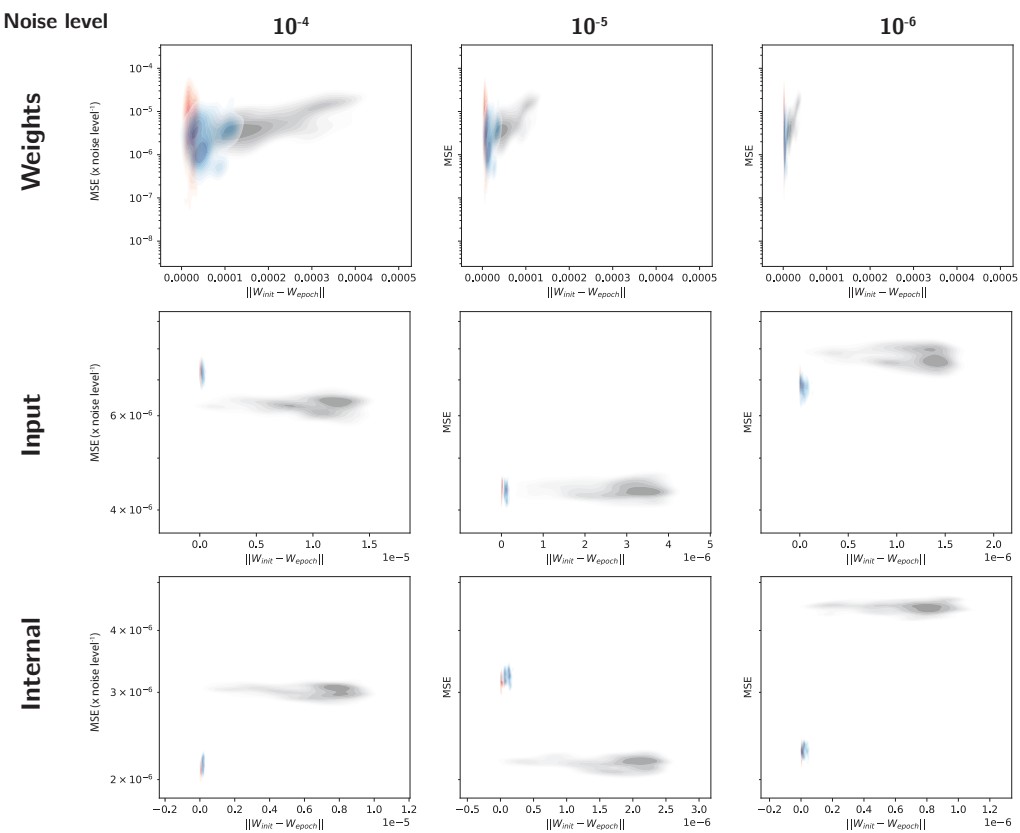

Figure S11: Distribution of parameters for the three noise types for three noise levels. Because of relative scale of perturbation, iRNN is further away from the initial parameters with internal and input noise. Depending on the level of the noise it performs better or worse than the LAs.

## S8    CHANGES TO THE NEURAL INTEGRATORS DURING LEARNING

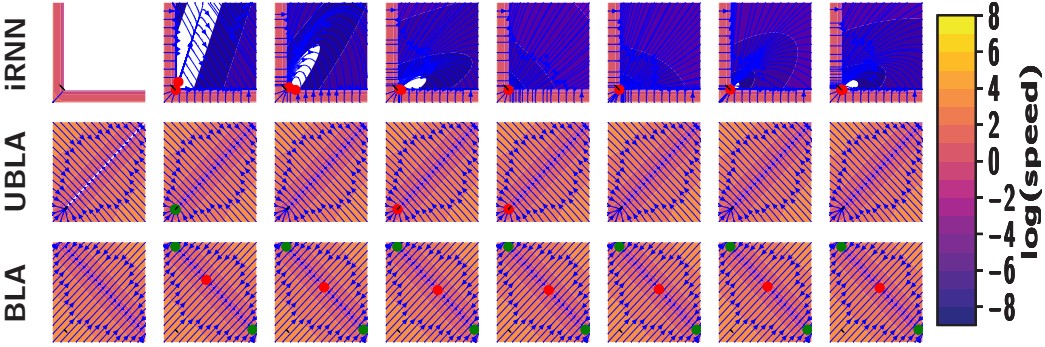

Figure S12: The dynamics of the recurrent part of the integrators across learning with some example orbits (blue lines), stable (green) and unstable (red) fixed points. A gradient step is taken after every perturbation. Gradient steps 0, 1, 5, 10, 15, 20, 25 and 29 are shown.

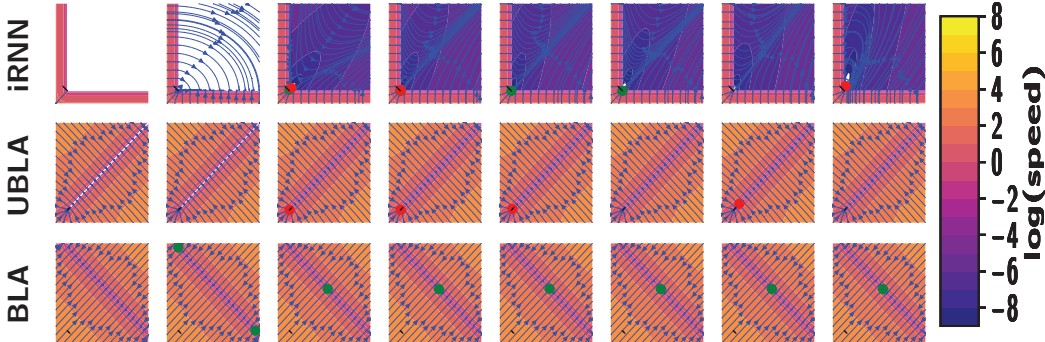

Figure S13: The dynamics of the recurrent part of the integrators across learning with some example orbits (blue lines), stable (green) and unstable (red) fixed points. A gradient step is taken after every 5 perturbations. Gradient steps 0, 1, 5, 10, 15, 20, 25 and 29 are shown.

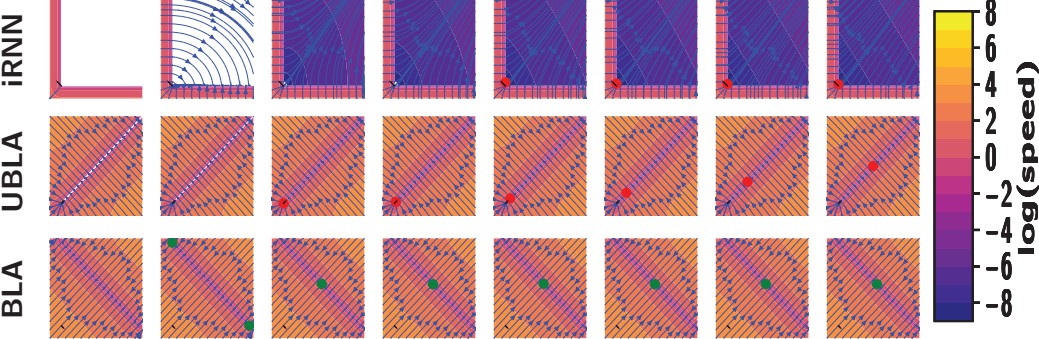

Figure S14: The dynamics of the recurrent part of the integrators across learning with some example orbits (blue lines), stable (green) and unstable (red) fixed points. 30 gradient steps are take after a single perturbation. Gradient steps 0, 1, 5, 10, 15, 20, 25 and 29 are shown.

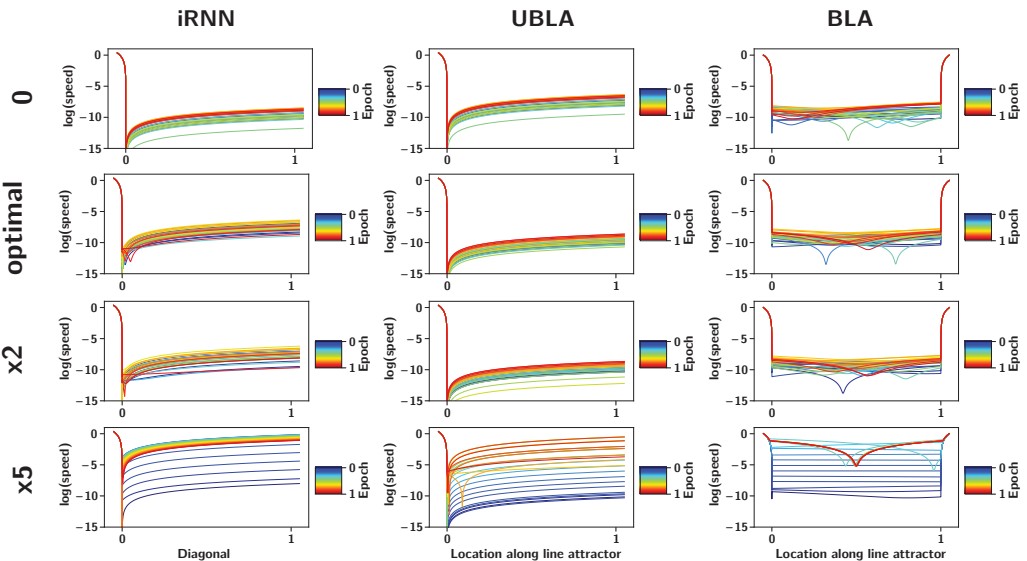

Figure S15: Speed along the invariant manifold during learning. For the iRNN a slice (the diagonal) is shown.

## S9 RING PERTURBATIONS

We define a local perturbation (i.e., a change to the ODE with compact support) through the bump function $\Psi(x) = \exp\left(\frac{1}{\|x\|^2 - 1}\right)$ for $\|x\| < 1$ and zero outside, by multiplying it with a uniform, unidirectional vector field. All such perturbations leave at least a part of the continuous attractor intact and preserve the invariant manifold, i.e. the parts where the fixed points disappear a slow flow appears. The parametrized perturbations are characterized as the addition of random matrix to the connection matrix.

## S10 HEADING DIRECTION NETWORK

$$\tau \dot{h}_j = -h_j + \frac{1}{N} \sum_k (W_{jk}^{sym} + v_{in} W_{jk}^{asym})\phi(h_k) + c_{ff}, j = 1, \ldots, N, \tag{28}$$

In the absence of an input ($v_{in} = 0$) fixed points of the system can be found analytically by considering all submatrices $W_\sigma^{sym}$ for all subsets $\{\sigma \subset [n]\}$ with $[n] = \{1, \ldots, N\}$. A fixed point $x^*$ needs to satisfy

$$x^* = -(W_\sigma^{sym})^{-1} c_{ff} \tag{29}$$

and

$$x_i^* < 0 \text{ for } i \in \sigma. \tag{30}$$

### S10.1 MEASURE ZERO CO-DIMENSION 1 BIFURCATIONS

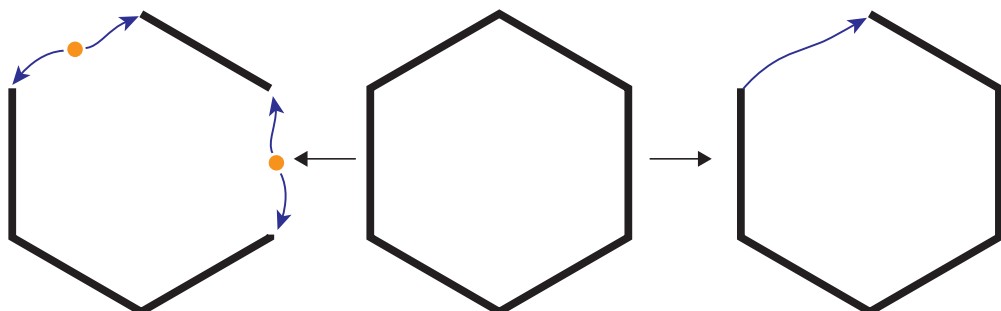

Figure S16: Measure zero co-dimension 1 bifurcations of the ring attractor network (Noorman et al., 2022).

### S10.2 INDEPENDENCE OF NORM OF PERTURBATION ON BIFURCATION

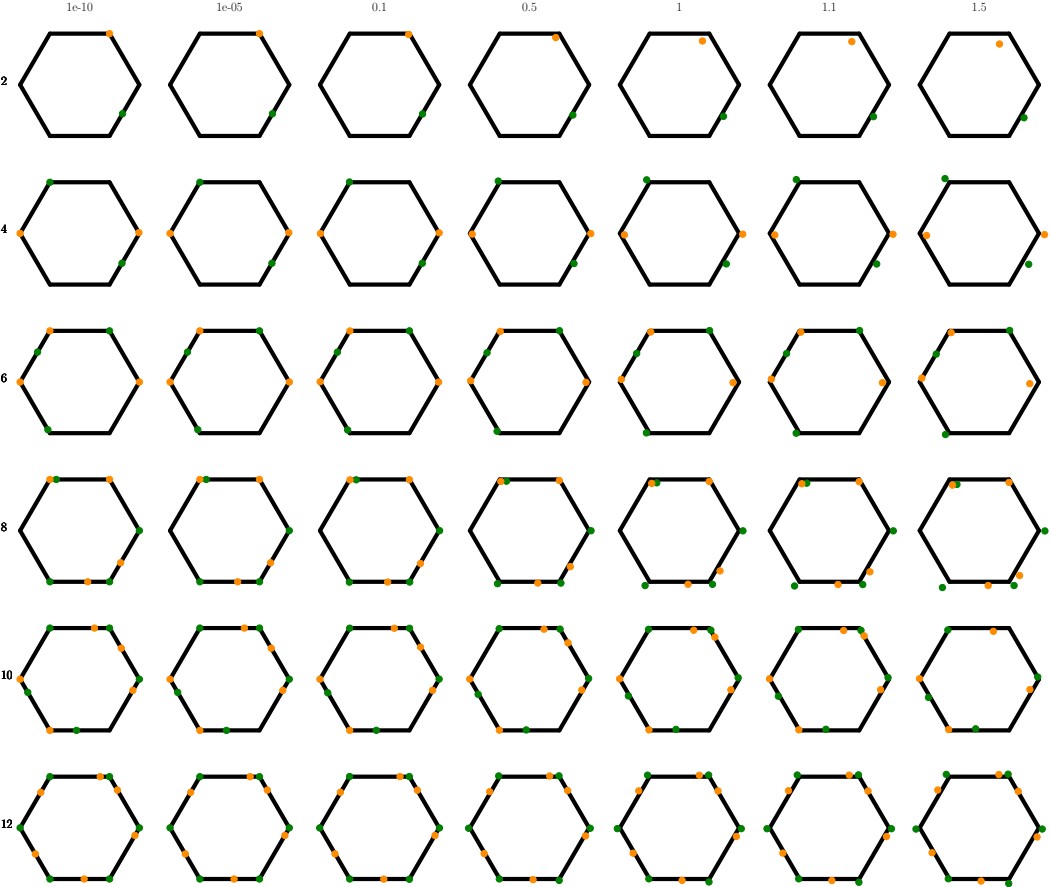

Figure S17: Rows show the bifurcations resulting from perturbations from the matrices with the same direction in Fig. 4A but with different norms (columns).

## S11    TRAINING RNNs ON AN INTEGRATION TASK FROM SCRATCH

We trained vanilla RNNs with a ReLU nonlinearity for the recurrent layer and a linear output layer on the angular velocity integration task Fig. S19. The network size varies between 50 and 200 units, initialized using a normal distribution for the parameters. Adam optimization with $\beta_1 = 0.9$ and $\beta_2 = 0.99$ was employed with a batch size of 512 and training was run until no decrease in the loss occurred for 100 epochs. The task has 256 time steps for the training samples, and training was based on the mean squared error loss.

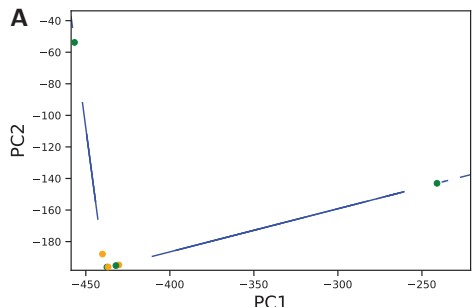 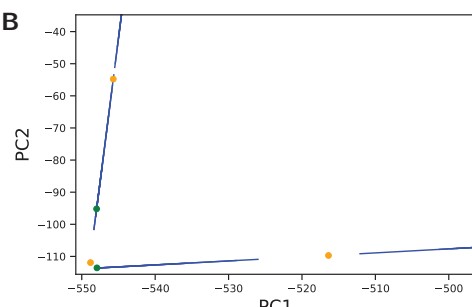

Figure S18: The two types of found solutions. A) A line attractor with hyperbolically stable fixed points at the end of the line. B) Saddle nodes at the ends of the line.

From an arbitrary initialization, we find that a line attractor-like structure often (8 out of 10 runs) emerged with hyperbolically stable fixed points (Fig.S18A) when trained on a longer version of the task. For shorter trial lengths, saddle nodes are more likely to emerge (6 out of 10 runs) at the ends of the line (Fig. S18B), meaning that the resulting structure is not an attractor.

## S12 TRAINING RNNS ON AN ANGULAR VELOCITY INTEGRATION TASK FROM SCRATCH

We trained vanilla RNNs with a tanh nonlinearity for the recurrent layer and a linear output layer on the angular velocity integration task Fig. S19. The network size is 20 units, initialized using a normal distribution for the parameters. Adam optimization with $\beta_1 = 0.9$ and $\beta_2 = 0.99$ was employed with a batch size of 512 and training was run until no decrease in the loss occurred for 100 epochs. The task has 100 time steps for the training samples, and the mean squared error loss was used.

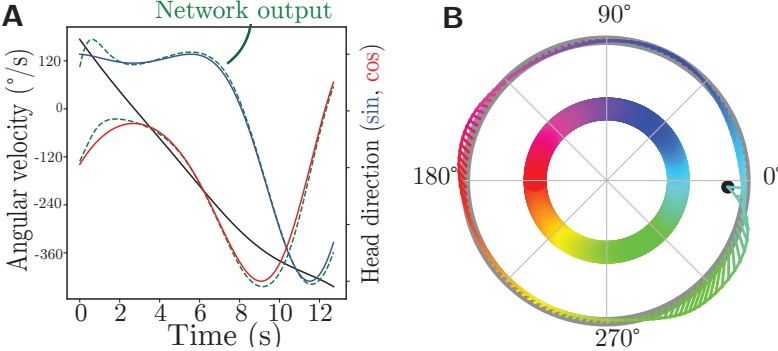

Figure S19: Description of the tasks. A) The angular velocity integration task. B) The output of the angular velocity integration in the output space, color coded according to the integrated angle. An example of an input is shown with constant velocity and it is provided until one turn is completed.

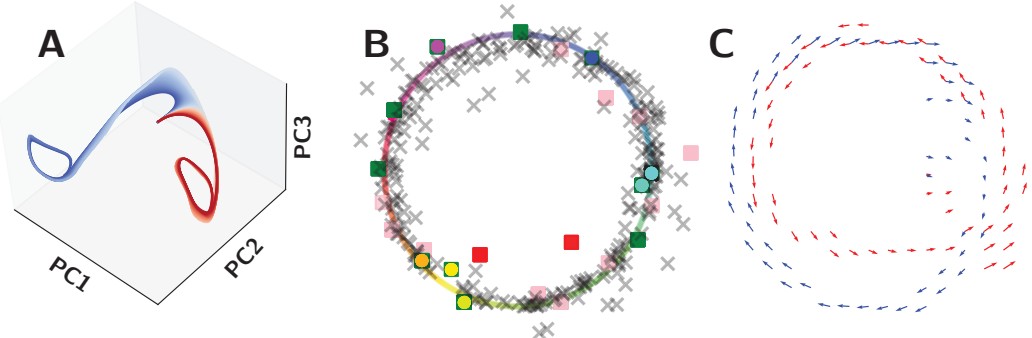

Figure S20: Analysis steps for the distillation of the implemented computation in a trained RNN. A) Input driven hidden trajectories for constant inputs of different magnitudes in the left (blue) and right (red) direction. B) Projection onto the output space of the attractors found by simulation until convergence to periodic solutions (color indicates the angular direction it maps to) with slow points found by minimizing the speed of the hidden (square) or output (cross) dynamics. Stability is indicated by green for stable, pink and red for saddles with 1 and 2 unstable dimensions, respectively. C) Effective input drive shown as average vector fields for the hidden dynamics projected onto the output space. Averages taken for a single constant input in left (blue) and right (red) directions.

For the angular velocity integration task, typical solutions have two limit cycles corresponding to the two directions of constant inputs. The autonomous dynamics can be characterized by an (approximate) line attractor with two (approximate) ring attractors at the ends. The found solutions Fig.S21A-C all show bounded ring-like attractors. These solutions are all composed of two rings (Fig.S20A) connected by an (approximate) line attractor.

The vector field (Fig. S20C) suggests that the system exhibits input driven dynamics corresponding to a limit cycle, which would mean that the invariant manifold of the input-driven dynamics is compact.

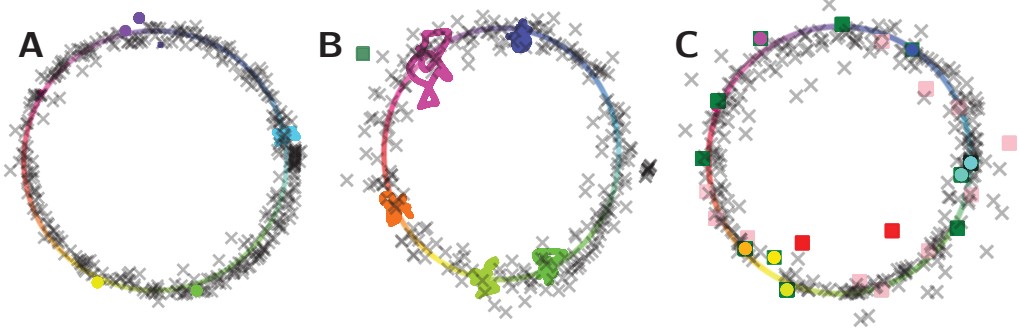

Figure S21: A) A solution with a single limit cycle (light blue) that gets mapped onto a small subset of the output space. B) A solution with multiple limit cycles spread around the ring attractor. C) A solution with only fixed points spread around a ring like attractor with slow dynamics.

