# OpenReview forum: "RNNS with gracefully degrading continuous attractors"
_ICLR.cc/2024/Conference — Submitted to ICLR 2024_

### Official Review · Reviewer_2ZH1 · 2023-10-26

**Soundness:** 2 fair
**Presentation:** 2 fair
**Contribution:** 2 fair
**Rating:** 3
**Confidence:** 2

**Summary:**

The focus of the present paper is on attractor networks, a special kind of recurrent neural network whose dynamics converge to a fixed point, or more generally towards a stable pattern within a fixed manifold. While attractor networks are prevalent tools in neurosciences and more generally in machine learning owing to their memory and computational properties, they suffer from instability upon changing one or several of their parameters, which impedes gradient-based learning in such models. The present paper proposes an existing theoretical framework to systematically address this problem (invariant continuous attractor manifold theory) and proposes a set of simple experiments to validate the relevance of this framework. More precisely:

- The introduction introduces a simple class of RNNs (Eq. 1) of which two classes can be derived in dimension 2: the UBLA model and the BLA model. The UBLA model exhibits a half line in the positive quadrant of $\mathbb{R}^2$ as a stable manifold (i.e. continuously made up of fixed points) while the BLA model has a bounded line in the same quadrant as a stable manifold. The stable manifold of the former can break upon changing the weights in some way, while the stable manifold of the latter model is more stable. This poses the question of how stable learning (in the sense of preserving the topology of these stable manifolds) can be achieved in these models.

- Section 2.1 introduces Theorem 1, an existing theorem which, heuristically speaking, states that if the stable / invariant manifold of the model is initially bounded and smooth, then it may not change upon small perturbations. The slightly perturbed manifold is called a "slow manifold". In the light of this theorem, we may understand why UBLA (whose stable manifold is unbounded) is unstable upon weight nudging, while BLA is.

- Section 2.2 discusses implications on ML, and in particular that models not satisfying Theorem 1 (e.g. UBLA) exhibit exploding gradients. For models satisfying Theorem 1 (e.g. BLA), it is suggested that loss gradients could act as "restorative forces" to counteract the deformation of the stable manifold. This idea is empirically investigated later in the paper.

- Section 2.3 discusses implication on neurosciences.

- Section 3.1 presents the first experiment which is the linear temporal integration task. In this setting, the RNN processes a sequence of pair of stochastic clicks and is learnt to predict the time-cumulated difference between the clicks. Therefore the RNNs under investigation operate in two dimensions and three kinds of RNNs are studied: the identity RNN (Id), and the UBLA and BLA model previously introduced. Fig. 3, along with the text, states the main findings. 1/ when deterministically perturbing a single entry of the recurrent weight matrix of these models, UBLA and Id can diverge (i.e. the loss diverges upon running the temporal dynamics) while BLA remains stable (B), 2/ when starting from the model optimum and adding a single noisy weight perturbation to the models, gradient descent maintains the model near optimum (E), yet often to a suboptimal solution (F), 3/ BLA exhibits some advantages in terms of stability, 4/ "in practice, it is difficult to maintain any of the continuous attractors".

- Section 3.2 presents similar experiments but in polar coordinates, with the same integration task but on this circle. Two models are considered, one whose stable manifolds are the origin and the unit circle, another one where the stable manifold is boundary less. However, results aren't presented on this model.

- The conclusion is somewhat negative : 1/ "the homeostatic restoration of continuous attractor [through gradient signal] may be challenging", 2/ "further researcher on finding restorative learning for larger perturbations in the parameter space is needed".

**Strengths:**

- The paper tackles a difficult problem
- An existing theoretical framework is proposed to account for some observations.

**Weaknesses:**

- The theoretical framework proposed, while enlightening the UBLA and BLA models, does not lead to a concrete prescription as to how to build RNNs that are robust to weight perturbations.
- The models considered are two-dimensional, how do the findings extend to larger models?
- If I understood the paper correctly, the sole takeaways of the study to have robust RNNs are to pick ones which have bounded and smooth stable manifolds and to use error gradients to counter act noise (however not so efficiently per the results).
- The experimental setup may not be relevant to larger models, given that the RNN robustness is studied when the model is *already initialized at its optimum*, which may not be the case for a realistic task.
- The way the generality of the results are phrased in the introduction ("these theoretical results **nullify** the concern of the fine tuning problem in neurosciences and suggest **general principles** for evaluating and designing new architectures and initialization strategies for RNNs in ML") starkly contrast with the experimental setup considered (namely 2-dimensional RNNs on toy problems) and the conclusion which tends to be negative. Also, I do not see in this paper any proposal of "initialization strategies for RNNs in ML".

**Questions:**

I must say upfront that I am not an expert of this literature, so it might be that the paper meets the standard of its literature to gain acceptance. This being said: could you please phrase in a clear fashion the concrete outcomes of your study for a general, high-dimensional RNN?

---

> ### Author Response · Authors · 2023-11-22
> **Response to Reviewer 2ZH1**
>
> We thank reviewer 2ZH1for the review and the valuable suggestions. Our responses to the raised points are the following:
>
> > The theoretical framework proposed, while enlightening the UBLA and BLA models, does not lead to a concrete prescription as to how to build RNNs that are robust to weight perturbations.
>
> There is some literature on the necessary conditions for attractive and bounded solutions to exist, see for example Yi (2001). See conditions for a line attractors in ReLU RNNs in Yu (2009), one just needs to rotate these line attractors so that they become bounded. Furthermore, conditions of the equilibria in more generality (line and plane attractors) of the 3D RELU network are presented in Xiang (2021) and parameters for bounded plane attractors for 4D tanh networks are discussed in Yu (2023). These construction should lead to RNNs with continuous attractors that are robust to weight perturbations given our theory.
>
>
> > how do the findings extend to larger models?
>
> The theory tells us that regardless of the dimensionality of the RNN, as long as they have bounded normally hyperbolic continuous attractors the only way to create exploding gradients is if there is a chaotic system in the neighborhood of the original.
> Currently, all systems considered have a slow flow after perturbation, which can be maintained with the correct learning rate. If one would take non normally hyperbolic attractors, the exploding gradients would form an even bigger problem. For RNNs with a ReLU  activation function, all continuous attractors are normally hyperbolic however, whether they are bounded or not.
>
> >to have robust RNNs are to pick ones which have bounded and smooth stable manifolds
>
> This indeed underlies the generality of the theory.
> However, it is not required that the invariant manifold be smooth.
> Furthermore, using backpropagated gradients would be one way to counteract noise, which is especially problematic for non-normally hyperbolic invariant manifolds.
>
> > The experimental setup may not be relevant to larger models, given that the RNN robustness is studied when the model is already initialized at its optimum, which may not be the case for a realistic task.
>
> We agree that our theory only covers cases where there is a continuous attractor implemented in an RNN. However, our theory does apply to RNNs of all sizes, hence also for larger models.
> these theoretical results nullify the concern of the fine tuning problem in neurosciences
> We agree with the reviewer that this is too strong of a claim given our current understanding of continuous attractors and we have softened our language in the manuscript accordingly.
> However, we would still like to note that the BLA has the broadest range of learning rates that are optimal and far away from really badly exploding gradients.
> Furthermore, the somewhat negative results are specific for the backpropagated gradient that we consider here. It is not a general principle that homeostatic restoration of continuous attractors is challenging, but only for the gradient signal considered in the paper. In fact, a control mechanism could be theoretically devised that relies on finite control input for all bounded continuous attractors, while this is not guaranteed for unbounded continuous attractors.
>
> > initialization strategies for RNNs in ML
>
> Further research needed to test efficiency of BLA as an initialization strategy. Our theory suggests that networks with bounded continuous attractors) will not exhibit exploding gradients under noisy conditions, while RNNs initialized with the identity matrix will.
>
>
> Thank you for your time and consideration.
>
>
>
> Yi, Z., Tan, K.K. and Lee, T.H., 2003. Multistability analysis for recurrent neural networks with unsaturating piecewise linear transfer functions. Neural Computation, 15(3), pp.639-662.
>
> Yu, J., Yi, Z., & Zhang, L. (2009). Representations of continuous attractors of recurrent neural networks. IEEE transactions on neural networks, 20(2), 368-372.
>
> Xiang, W., Yu, J., Yi, Z., Wang, C., Gao, Q., & Liao, Y. (2021). Coexistence of continuous attractors with different dimensions for neural networks. Neurocomputing, 429, 25-32.
>
> Yu, J., Chen, W., Leng, J., Wang, C., & Yi, Z. (2023). Weight matrix as a switch between line attractor and plane attractor of ring neural networks. Neurocomputing, 521, 181-188.

---

### Official Review · Reviewer_EwtU · 2023-11-05

**Soundness:** 3 good
**Presentation:** 3 good
**Contribution:** 2 fair
**Rating:** 5
**Confidence:** 3

**Summary:**

This paper analyzes the sensitivity of the attractors learned by RNNs to changes in its parameters. This analysis is based on Fenichel duality -- informally, decomposition of the phase space into slow and fast manifolds -- and Fenichel's theorem that gives the robustness of the slow manifold for certain bifurcations. They hypothesize that, in two dimensions, RNNs that have exploding gradients encounter bifurcations where Fenichel's theorem does not apply. This is illustrated with numerical examples.

**Strengths:**

The illustrative example and visualizations in two dimensions are quite useful to understand the overall message.

**Weaknesses:**

The biggest concerns are i) the lack of mathematically precise explanations and ii) the lack of guidance as to how to interpret the 2D results for RNN training in general. Specifically, it would be better if the implications of Fenichel's theorem for RNN training can be discussed more precisely. Can we prove that satisfying the conditions of the theorem is sufficient for convergence of training?

There is a brief remark about bifurcations into chaos also causing exploding gradients. Does this mean that the attractors of the hidden states are never chaotic attractors?

While the authors mention that these theoretical insights can be used to guide RNN training dynamics away from problematic bifurcations, it is unclear how that can be accomplished in practice, in high-dimensions.


Minor:
1. Notation $\mathcal{M}_0$ and $M_0$ not used consistently.
2. Grammatical errors, e.g. " The BLA has a bounded loss in *it’s* neighborhood. "

**Questions:**

Apart from the questions in the previous section, it would be useful to answer some others that clarify the scope of the results. When can RNN dynamics be decomposed into slow-fast systems for which we can apply Fenichel duality? When does the input dynamics (discrete-time map) induce Fenichel-type attractors in the dynamics of the hidden state (may be the authors can expand a bit on their observations about the loss landscape near the normally hyperbolic fixed point).

Even chaotic attractors can be structurally stable (e.g., Anosov systems or uniformly hyperbolic systems). So, clearly the lack of structural stability is not sufficient for keeping the RNN training dynamics from developing exploding gradients. Hence, what are the necessary and sufficient conditions for stable training dynamics?

---

> ### Author Response · Authors · 2023-11-22
> **Response to Reviewer EwtU**
>
> We thank reviewer EwtU for the review and the valuable suggestions. Our responses to the raised points are the following:
>
> > i. the lack of mathematically precise explanations
>
> We have adjusted the language in Theorem 1 (Fenichel's Invariant Manifold Theorem) to make the statement mathematically mode precise. To comment on the complications of determining the extent of noise that a given system can handle, a lot more research is needed to develop a full account for what the maximal $\epsilon$ (the timescale parameter) value is for which persistence holds. But in principle it can be calculated, see Chapter 6.5 in Wiggins (1994).
>  If there are any particular parts of the manuscript that the reviewer  found imprecise, we would be happy to know so that we can improve the overall mathematical precision.
>
> > ii. the lack of guidance as to how to interpret the 2D results for RNN training in general
>
> Normal hyperbolic attractors are persistent in arbitrary dimensions. This is demonstrated for a higher dimensional example of the head direction network implementing a ring attractor. All perturbations in the vicinity of the original system lead to a bounded invariant manifold, for which we describe the Morse decomposition in terms of the fixed points and their connecting orbits. Saddle nodes are always connected to stable fixed points in the direction of their unstable manifold.
>
> > Does this mean that the attractors of the hidden states are never chaotic attractors?
>
> For 1 and 2 dimensional continuous attractors, small enough perturbations never lead to chaotic attractors, because the perturbed invariant manifold needs to be homeomorphic to the original continuous attractor and there can be no chaos in dimensions 1 or 2 (see for example the Poincaré–Bendixson theorem). For higher dimensional continuous attractors this is not guaranteed, however, the chaotic flow will be slow according to the theory.
>
> > While the authors mention that these theoretical insights can be used to guide RNN training dynamics away from problematic bifurcations, it is unclear how that can be accomplished in practice, in high-dimensions.
>
> Normal hyperbolic attractors will be persistent in arbitrary dimensions, even in PDEs. For ReLU RNNs, there is some literature on the necessary conditions for attractive and bounded solutions to exist, see for example Yi (2001). See conditions for a line attractors in linear threshold RNNs in Yu (2009), one just needs to rotate these line attractors so that they become bounded. Furthermore, conditions of the equilibria in more generality (line and plane attractors) of the 3D RELU network are presented in Xiang (2021) and parameters for bounded plane attractors for 4D tanh networks are discussed in Yu (2023).
>
> We will include these in the manuscript. However, for RNNs with arbitrary activation functions, a general theory is still lacking.
>
> > When can RNN dynamics be decomposed into slow-fast systems for which we can apply Fenichel duality?
>
> For continuous attractors this decomposition is always possible. We would kindly bring to the attention of the reviewer that Fenichel’s Invariant Manifold Theorem should not to be confused with Fenchel’s duality theorem (https://en.wikipedia.org/wiki/Fenchel%27s_duality_theorem).
> For other systems, decomposability into fast and slow subsystems needs to be investigated on a case-by-case basis.
>
> > When does the input dynamics (discrete-time map) induce Fenichel-type attractors in the dynamics of the hidden state (may be the authors can expand a bit on their observations about the loss landscape near the normally hyperbolic fixed point).
>
> We show some results of RNN trained on an integration task that show that bounded normally hyperbolic invariant sets (is this what the reviewer refers to as Fenichel-type attractors?) develop during learning and that even with (constant) input the system exhibits Fenichel-type attractor behavior, typically in the form of limit cycles and fixed points. These results are included in Supp.Sec. 12 (Supp.Fig. 20.
>
> Thank you for your time and consideration.
>
>
> Yi, Z., Tan, K.K. and Lee, T.H., 2003. Multistability analysis for recurrent neural networks with unsaturating piecewise linear transfer functions. Neural Computation, 15(3), pp.639-662.
>
> Yu, J., Yi, Z., & Zhang, L. (2009). Representations of continuous attractors of recurrent neural networks. IEEE transactions on neural networks, 20(2), 368-372.
>
> Xiang, W., Yu, J., Yi, Z., Wang, C., Gao, Q., & Liao, Y. (2021). Coexistence of continuous attractors with different dimensions for neural networks. Neurocomputing, 429, 25-32.
>
> Yu, J., Chen, W., Leng, J., Wang, C., & Yi, Z. (2023). Weight matrix as a switch between line attractor and plane attractor of ring neural networks. Neurocomputing, 521, 181-188.

---

### Official Review · Reviewer_yzhP · 2023-11-05

**Soundness:** 3 good
**Presentation:** 2 fair
**Contribution:** 2 fair
**Rating:** 5
**Confidence:** 4

**Summary:**

The authors utilized Fenichel’s persistence theorem to demonstrate that bounded line attractors are stable, meaning they maintain stability even when subjected to perturbations. This stability ensures that, when a corrective learning signal is present, backpropagation will not result in exploding gradients over any duration. In contrast, unbounded line attractors can become divergent systems when subjected to specific perturbations, leading to exploding gradients. This observation also suggests that specific implementations of continuous attractors may have built-in homeostatic mechanisms that preserve the attractor's structure, making it suitable for the neural computations it is intended for.

**Strengths:**

- The paper is well-written and its main idea is interesting. Especially, I liked the idea of using Fenichel’s invariant manifold theorem to address the discussed issue.

- The visualizations provided in Fig. 1 and 2 are very helpful in providing a clear illustration of their method.

**Weaknesses:**

1. First off, it's important to note that, as demonstrated in the study by Mikhaeil et al. (2022), when dealing with chaotic or unstable dynamics, gradients will always explode and **this issue cannot be solved by RNN architectural design or regularization and needs to be addressed in the training process**. So, the authors should consider this more precisely.

2. The paper primarily focuses on a specific RNN variant, employing the ReLU activation function, 2-dimensional RNNs with specific parameter values outlined in equations (9) and (10). This specificity may limit the generalizability of its findings to higher dimensions, other architectural designs, and parameter configurations. In one particular parameter setting, the RNN exhibits 'well-behaved' dynamics, converging to a fixed point with bounded loss gradients. In such cases, established solutions (e.g., Hochreiter & Schmidhuber 1997; Schmidt et al., 2021) can be employed to prevent gradient vanishing. While this result is interesting, it's essential to recognize that it represents only one specific scenario and may not cover the broader spectrum of possibilities.

3. Utilizing Fenichel’s invariant manifold theorem, the authors established a sufficient condition to ensure that RNNs implementing continuous attractors remain immune to exploding gradients. However, pleae note that a ReLU RNN, discrete- or continuous-time, is a piesewise linear (PWL) dynamical system, a subclass of piesewise smooth (PWS) dynamical systems, that is smooth everywhere except on some boundaries separating regions of smooth behavior. These boundaries, called switching manifolds or borders, divide the phase space into countably many regions. Many existing theories of smooth DS do not extend to PWS systems. In PWS systems the interaction of invariant sets with switching manifolds often gives rise to bifurcations, known as discontinuity-induced bifurcations, which are not observed in smooth systems. Moreover, in the PWS setting, for bifurcations containing non-hyperbolic fixed points, similar slow (center) manifold approaches can be applied, but only to part of phase space. However, bifurcations involving switching manifolds cannot be examined in the same manner, as there is no slow (center) manifold. Consequently, the applicability of Fenichel’s invariant manifold theorem to different parameter settings of the considered system (eq. (7)) may be limited or pose considerable challenges.

4. Regarding the link between bifurcations and EVGP during training, it is essential to consider additional relevant literature. For instance, as demonstrated in (https://arxiv.org/pdf/2310.17561.pdf) specific bifurcations in ReLU-based RNNs are always associated with EVGP during training.

**Questions:**

Can the results of this paper be generalized to more general settings?

---

> ### Author Response · Authors · 2023-11-22
> **Response to Reviewer yzhP**
>
> We thank reviewer yzhP for the review and the valuable suggestions. Our responses to the raised points are the following:
>
> 1. In this paper we focus on the properties of continuous attractors that might be necessary for some specific tasks, and are not concerned with learning chaotic dynamics.
> 	Furthermore, unstable dynamics won’t be present in our proposed RNNs as we are concerned with bounded continuous attractors that will have a persistent attractive invariant manifold. We clarified this in the paper on page 3 at the beginning of Section 2.
>
> 2. We agree that the paper only covers one specific scenario (that of bounded invariant manifolds in RNNs) and may not cover the broader spectrum of possibilities. However, note that our theory is applicable to all RNNs and biological recurrent networks that can realize such invariant manifold structures. We believe that it is this generality of the theory that makes it relevant to the broader theoretical neuroscience and machine learning communities.
>
> 3. Smoothness of the system under consideration is indeed a valid worry in general. However, for continuous piecewise linear (PWL) dynamical systems for our purposes of stability of the persistent invariant manifold, the theory is applied (ReLU RNNs are continuous). If the critical manifold is globally stable then the system is forward invariant in a neighbourhood of the critical manifold, and Fenichel’s invariant manifold theorem is applicable (Simons, 2018). We believe that the theory can be applied to PWL continuous systems with continuous attractors in high generality because of their global attraction property, see for example the analysis of the head direction model in Section 3.2.
> For remarks on the application of the fast-slow theory to PWLs, see Theorem 19.3.3 in Kuehn (2015). And in practice we see that the continuous attractor persists as an invariant manifold, see the bifurcation analysis both on the bounded line attractor network and the head direction representation network. For Filippov systems the situation is different because they are not continuous. For fast-slow analysis of Filippov systems, see Cardin (2013), however, note that these are outside the scope of our investigation. We included a remark on the applicability of Fenichel’s invariant manifold theorem to continuous piecewise linear (PWL) dynamical systems on page 4 in the second paragraph.
> This means that even for PWL continuous dynamical systems, the normally hyperbolic invariant manifold that is the continuous attractor is going to persist. It will not maintain smoothness because $r<1$, but we don’t think that causes an issue for the (biological) systems that implement continuous attractors. If it is important that the invariant manifold persists in a smooth way, namely that the perturbed invariant manifold is $C^r$ then other activation functions need to be used.
>
>
>
>
> 4. We thank the reviewer for this suggestion and will incorporate it in the text as one way of finding maximal epsilon bifurcation parameters around continuous attractors.
> Our theory shows that for 1 and 2 dimensional continuous attractors, there are no exploding gradients for small enough perturbations, even though bifurcations do occur. The only possibility is the creation of chaotic orbits, which could still cause a problem. This paper however is not considering bifurcations involving chaotic behavior, so we don’t see the relevance of considering this part of the literature.  Note finally that while we acknowledge the relevance of this recent paper to our research, it was published after the submission of our manuscript.
>
> Thank you for your time and consideration.
>
> Cardin, P., Da Silva, P., & Teixeira, M. (2013). On singularly perturbed Filippov systems. European Journal of Applied Mathematics, 24(6), 835-856.
>
> Kuehn, C. (2015). Multiple time scale dynamics (Vol. 191). Berlin: Springer.
>
> Simpson, D.J. (2018). Dimension reduction for slow-fast, piecewise-smooth, continuous systems of ODEs. arXiv preprint arXiv:1801.04653.

---

### Official Review · Reviewer_K1Kz · 2023-11-06

**Soundness:** 2 fair
**Presentation:** 3 good
**Contribution:** 2 fair
**Rating:** 5
**Confidence:** 4

**Summary:**

This paper uses the invariant manifold theorem and classifies manifolds into those that are bounded (and obey the invariant manifold theorem), and those that are unbounded (which don’t obey the invariant manifold theorem). They then show, consistent with the invariant manifold theorem, that unbounded attractors are more resistant to perturbation of RNN weight matrices. They connect this to exploding gradients in RNNs, and show that the bounded attractors have lower gradients than the others.

**Strengths:**

I like the aims of this paper. It is (mostly) clearly presented, and is original to me at least. I am concerned however about the generality of the approach (see below), and my take is that the idea and subsequent testing is a bit too preliminary to make any real conclusions.

**Weaknesses:**

Only two very simple settings are considered, and so it’s hard to know how general this approach is. The only learning considered is learning at small displacements from prespecified attractors. I would be nice to see are the bounded attractors more likely to be learned from arbitrary initialisations. What happens in complicated situations? Can any insights be made into the sorts of sequence tasks that machine learners care about? What does it mean for biological networks? Are there learning algorithms that lead to these bounded attractors?

Fig 3 is so cluttered it is hard to parse.

**Questions:**

See weaknesses

---

> ### Author Response · Authors · 2023-11-22
> **Response to Reviewer K1Kz**
>
> We thank reviewer K1Kz for the review and the valuable suggestions.
>
> Before addressing the comments of the reviewer, we would like to make a statement on the generality of the approach.
> While we acknowledge the importance of practical validation, we firmly believe that our work holds substantial merit even in the absence of these experiments. Allow us to elaborate on the theoretical progress we aim to contribute to the field. Our study focuses on a fundamental theoretical insight that transcends the immediate realm of practical applications. We establish that, under the condition of attractors being normally hyperbolic, they exhibit a robustness to small perturbations, thereby retaining their attractiveness as a set. This finding not only adds a valuable theoretical dimension to our understanding of machine learning but also has broader implications for the advancement of the field.
>
> In essence, our work strives to unveil general principles that govern the stability of attractors, contributing to the foundational knowledge of machine learning dynamics. While we acknowledge that practical applications may not be readily implementable, we argue that the theoretical framework we propose lays the groundwork for future innovations and discoveries.
>
> We invite the reviewer to consider the broader impact of our theoretical contributions, which extend beyond the confines of specific experiments. By elucidating the stability properties of attractors, our work opens avenues for further exploration and application in diverse contexts. We remain committed to advancing the field of machine learning and believe that our theoretical findings provide a solid foundation for future research endeavors.
>
> Our responses to the raised points are the following:
>
> > are the bounded attractors more likely to be learned from arbitrary initialisations
>
> We agree with the reviewer that bounded attractors should be easier to learn, however, only if the network dynamics is already near the solution. From an arbitrary initialization, we find that various slow manifold solutions that approximate a continuous attractor can be reached. In our numerical experiments, when circular variables were trained, a ring-like structure often emerged. We added these results to the supplementary material in Supp. Sec. 12.
> We furthermore present some preliminary results on training RNNs on the same integration task, but from arbitrary initialization for the parameters. In this case, we show that bounded attractors are more likely to be learned if the task duration is long enough.
>
>
>
> > What happens in complicated situations?
>
> We will try to clarify what could happen in an example of a task that is more complicated. We will reflect on RNNs trained on multiple tasks, such as in Driscoll (2022). In this case, one would need to study how certain invariant manifolds used by one task are influenced by training the system on another. One could then another and then to compare different implementations of a line attractor (UBLA/BLA) while training the system on two different tasks, an integration task and a perceptual decision-making task for example.
>
> In the supplementary, we now show some results on training RNNs on an angular velocity integration task (for which continuous,  circular variables need to be integrated and maintained by the network). In this case, the trained RNNs exhibit a ring-like structure.
>
> If the reviewer had other complicated situations in mind, we would be happy to address what happens in them.
>
> > What does it mean for biological networks?
>
> As suggested in the paper, structural constraints (such as Dale's law and mutual inhibition which are both satisfied for the bounded line attractor) could contribute to biasing learning towards bounded attractors.
>
>
> > Can any insights be made into the sorts of sequence tasks that machine learners care about?
>
> Our theory suggests that compact continuous attractors are more robust under noisy and continual learning. For high-dimensional RNN training for general tasks (e.g. sequential MNIST), as long as these involve the need to represent continuous variables, continuous attractors will be a natural solution, see for example Dehyadegary (2011).

---

> > ### Author Response · Authors · 2023-11-22
> > **Response to Reviewer K1Kz**
> >
> > > Are there learning algorithms that lead to these bounded attractors?
> >
> > This is an excellent question for which a new theoretical framework should be developed. This would however need to describe learning dynamics of training of RNNs, instead of robustness properties of continuous attractors of RNNs under noise, which is the focus of this paper. As suggested in the paper, structural constraints (such as Dale's law and mutual inhibition) could contribute to biasing learning towards bounded attractors.
> >
> >
> > Thank you for your time and consideration.
> >
> > Dehyadegary, L., Seyyedsalehi, S.A. and Nejadgholi, I. (2011). Nonlinear enhancement of noisy speech, using continuous attractor dynamics formed in recurrent neural networks. Neurocomputing, 74(17), pp.2716-2724.
> >
> > Driscoll, L., Shenoy, K., & Sussillo, D. (2022). Flexible multitask computation in recurrent networks utilizes shared dynamical motifs. bioRxiv, 2022-08.

---

> ### Comment · Reviewer_K1Kz · 2023-11-23
> **Thanks for your responses**
>
> Many thanks for your responses.
>
> I still think the result and potential avenues are interesting, but, while I appreciate the additional results, I still think that much more needs to be done to actually prove your points. I have nevertheless raised my score to reflect the addiional results.

---

### Author Response · Authors · 2023-11-22
**General comment on key clarifications and additional results**

We are grateful for the insightful and constructive feedback provided by the reviewers.

# Clarification on the primary contribution
We would like to reiterate the primary contribution to the specific problem of memory and learning over time. Our work addresses a universal concern rooted in the structural instability of continuous attractors. In general, continuous attractors are the only widely used mechanism for remembering continuous values over time–just like the tape of Turing machines, continuous attractors are central to the information manipulation during computation without quantization. It applies to specialized recurrent architectures such as LSTM and theoretical models of biological neural networks such as the head direction system and prefrontal cortex networks for working memory. Although ML practitioners may not be aware of these fundamental issues, our theory conceptually advances the understanding of internal representation in network activation patterns related to learning and memory.  We strongly argue that our line of research will lead to better architectures and learning systems in the future, and not every contribution at ICLR has to have immediate consequence to ML practitioners.

# Revisions
Thank you to all reviewers for their initial comments and ongoing discussion about our work. We have uploaded a preliminary revised version of our paper based on reviewer feedback.
The notable changes are as follows:

1. Adding a remark about what kind of smoothness is guaranteed in Theorem 1.
2. Changing "nullify the concern" to "significantly mitigate the concern" (top of page 3)
3. Addition of "while the BLA has the broadest range of learning rates that are optimal and far away from exploding gradients" in Section 3.1.4
4. Clarification of the focus of the paper on the properties of continuous attractors that might be necessary for some specific tasks, and are not concerned with learning chaotic dynamics. Furthermore, unstable dynamics won’t be present in our proposed RNNs as we are concerned with bounded continuous attractors that will have a persistent attractive invariant manifold. (on page 3 at the beginning of Section 2)
5. A remark on the applicability of Fenichel’s invariant manifold theorem to continuous piecewise linear (PWL) dynamical systems on page 4 in the second paragraph.
6. Figure 3 has been restructured
7. New experiments:
    1. Training RNNs on the continuous version of the Poisson integration task described in the manuscript (Sup.Sec 11). From an arbitrary initialization, we find that a line attractor-like structure often emerged with hyperbolically stable fixed points at the end of the line when trained on a longer version of the task. For shorter trial lengths, saddle nodes are more likely to emerge at the ends of the line, meaning that the resulting structure is not an attractor.

    2. Training RNNs on the angular velocity integration task  (Sup.Sec 12). From an arbitrary initialization, we find that a ring-like structure often emerged.




Once again, we extend our gratitude to the reviewers for their constructive feedback, which has undoubtedly enriched the depth and quality of our contribution. We look forward to further discussions and potential future collaborations to collectively propel the field forward.

---

### Meta-Review · Area_Chair_ifUD · 2023-12-06

**Metareview:**

This paper examines continuous attractors in recurrent neural networks (RNNs) with a focus on bounded line attractors (BLAs). The authors use Fenichel's persistence theorem to show that BLAs are stable, meaning that they will remain attractive under perturbations (whereas unbounded line attractors will not). As well, they connect this to gradient propagation, observing that when the theorem applies, gradients propagated through the network should not explode, whereas if it doesn't apply, they should. The authors verify their theories with experiments on 1 and 2-D continuous attractors.

The reviews for this paper were low-to-borderline. There were consistent concerns amongst the reviewers about the generality and practical usefulness of the results. For example, whether the theorem would hold for piecewise linear systems was raised, and it was noted that the results do not provide practical guides on how to take construct BLAs when training a RNN. The authors provided rebuttals to address some of these concerns, but the reviewers were not fully convinced, and the scores remained at 5,5,5,3. Taking into consideration that two reviewers did not respond to the rebuttal, it is the AC's opinion still that the concerns about practicality for ML were not fully addressed in the rebuttal. Given this, and the relatively low scores, a decision of 'reject' was reached. But, it should be noted that there are interesting aspects to this paper, and its contributions may be very well suited to another venue.

**Justification For Why Not Higher Score:**

This paper's practical relevance is unclear - it is mostly a nice explanation for a certain class of RNNs' behaviors. Given this, and the relatively low scores, I don't think accepting this paper would be appropriate.

**Justification For Why Not Lower Score:**

N/A

---

### Decision · Program_Chairs · 2024-01-16

Reject